# Oscillations support short latency co-firing of neurons during human episodic memory formation

**Frédéric Roux[1], George Parish[1], Ramesh Chelvarajah[1,2], David T Rollings[2], Vijay Sawlani[1,2], Hajo Hamer[3], Stephanie Gollwitzer[3], Gernot Kreiselmeyer[3], Marije J ter Wal[1], Luca Kolibius[4], Bernhard P Staresina[1,5], Maria Wimber[1,4], Matthew W Self[6], Simon Hanslmayr[1,4]***

[1]School of Psychology, Centre for Human Brain Health, University of Birmingham, Birmingham, United Kingdom; [2]Complex Epilepsy and Surgery Service, Neuroscience Department, Queen Elizabeth Hospital Birmingham, Birmingham, United Kingdom; [3]Epilepsy Center, Department of Neurology, University Hospital Erlangen, Erlangen, Germany; [4]School of Psychology and Neuroscience, Centre for Cognitive Neuroimaging, University of Glasgow, Glasgow, United Kingdom; [5]Department of Experimental Psychology, University of Oxford, Oxford, United Kingdom; [6]Department of Vision and Cognition, Netherlands Institute for Neuroscience, an institute of the Royal Netherlands Academy of Art and Sciences, Amsterdam, Netherlands

**\*For correspondence:**
simon.hanslmayr@glasgow.ac.uk

**Competing interest:** The authors declare that no competing interests exist.

**Abstract** Theta and gamma oscillations in the medial temporal lobe are suggested to play a critical role for human memory formation via establishing synchrony in neural assemblies. Arguably, such synchrony facilitates efficient information transfer between neurons and enhances synaptic plasticity, both of which benefit episodic memory formation. However, to date little evidence exists from humans that would provide direct evidence for such a specific role of theta and gamma oscillations for episodic memory formation. Here, we investigate how oscillations shape the temporal structure of neural firing during memory formation in the medial temporal lobe. We measured neural firing and local field potentials in human epilepsy patients via micro-wire electrode recordings to analyze whether brain oscillations are related to co-incidences of firing between neurons during successful and unsuccessful encoding of episodic memories. The results show that phase-coupling of neurons to faster theta and gamma oscillations correlates with co-firing at short latencies (~20–30 ms) and occurs during successful memory formation. Phase-coupling at slower oscillations in these same frequency bands, in contrast, correlates with longer co-firing latencies and occurs during memory failure. Thus, our findings suggest that neural oscillations play a role for the synchronization of neural firing in the medial temporal lobe during the encoding of episodic memories.

## Editor's evaluation

Roux and colleagues measured spiking activity and local field potentials predominantly from the hippocampus and also a few surrounding structures in the medial temporal lobe from patients with pharmacologically intractable epilepsy while the patients performed an associative memory task. Their data are convincing and provide important insights into how neurons in the medial temporal lobe correlate with associative memory.

## Introduction

Episodic memory relies on efficient information transmission within the medial temporal lobe (MTL; *Fell and Axmacher, 2011*; *Fell et al., 2001*; *Solomon et al., 2019*). More specifically, if one group of neurons drives neural discharges of another group of neurons, synaptic modifications can occur which transform fleeting experiences into durable memory traces (*Kandel, 2001*). The strengthening of synaptic connections between neurons that are active during the experience of an episode depends critically on the temporal structure of neural firing (*Markram et al., 1997*; *Bi and Poo, 2001*; *Hebb, 1949*; *Wespatat et al., 2004*). Evidence that has accumulated over several decades suggests that coordinated rhythmic activity may provide a candidate mechanism to establish fine-grained temporal structure on neural firing (*Buzsáki, 2010*; *Fries, 2015*; *Singer, 1999*). Accordingly, brain oscillations at theta (~3–9 Hz) and gamma (~40–80 Hz) frequencies in the human MTL have been proposed to promote the formation of memories through the synchronization of neural firing in the MTL (*Fell and Axmacher, 2011*; *Jutras and Buffalo, 2010*; *Huerta and Lisman, 1995*; *Hyman et al., 2003*).

Studies in animals showed that correlated neural firing is fundamentally involved in the strengthening of synaptic connections (*Hebb, 1949*). Consistent with a critical role of neural synchronization studies in humans using invasive and non-invasive electrophysiological recordings demonstrated that theta and gamma oscillations in the MTL are correlated with memory encoding (*Fell and Axmacher, 2011*; *Hanslmayr et al., 2016*). For instance, stronger phase-coupling of neural firing in theta (*Rutishauser et al., 2010*) oscillations has been observed during successful compared to unsuccessful encoding. A similar result has been obtained in macaques for the high gamma oscillations (>60 Hz; *Jutras et al., 2009*). Furthermore, increased cross-frequency coupling (CFC) between theta-phase and the power of gamma oscillations for successfully encoded memory items has also been reported in the human MTL (*Griffiths et al., 2021*; *Lega et al., 2016*). Therefore, it can be hypothesized that increased synchronization between neural spiking and theta and gamma oscillations is positively related to memory encoding, presumably via inducing efficient information transmission between neural ensembles.

However, efficient information transmission, and synaptic plasticity as a result thereof, may not only be reflected in the absolute level of synchronization, but also in the frequency at which this synchronization occurs. Neurons integrate input over time, with the rate of relaxation of the membrane potential dictating the length of the integration window (10–30 ms for neocortical principal cells; *Buzsáki, 2010*). Relatively faster oscillations integrate neural spikes over shorter time windows compared to relatively slower oscillations. One advantage of such a tighter temporal packing of spikes of an upstream neuron is that it is more likely to overcome the firing threshold of a downstream neuron because the individual spikes build on each other before the membrane potential fully drops back to baseline (*Vöröslakos et al., 2018*; *Geisler and Goldberg, 1966*). Therefore, a neural assembly which synchronizes firing at faster oscillations is more likely to drive a down-stream neural assembly compared to synchronization at slower oscillations. This may be the reason for why fast (~65 Hz), but not slow (~40 Hz), gamma oscillations in rodents (*Colgin et al., 2009*) and humans *Griffiths et al., 2019* have been demonstrated to reflect memory encoding processes.

This study aims to advance our understanding of how brain oscillations within the human MTL mediate neural firing in the service of episodic memory formation. Our understanding of this process is limited because previous studies have either investigated neural oscillations only in the local field potential or focused on single frequency bands (i.e. theta or gamma) or used simple recognition tasks which do not fully tap into the complex association processes underlying episodic memory. A recent study demonstrated a role of sleep spindles for modulating short-latency co-firing between neurons in the lateral anterior temporal cortex (*Dickey et al., 2021*), which is consistent with the idea that oscillations in principle can regulate efficient information transfer between neurons. However, little is known about the relevance of such co-firing between neurons or groups of neurons during human memory formation. This study aims to fill these gaps by simultaneously recording LFPs and neural firing in parallel from multiple micro-wire electrodes during an associative episodic memory task (*Figure 1A*). Our results show that successful memory formation is correlated with spike-LFP coupling at relatively faster theta and gamma oscillations as opposed to relatively slower oscillations. Furthermore, we show that gamma oscillations are coupled to the phase of theta oscillations specifically for successful memory trials, and that successful memory trials are characterized by short-latency

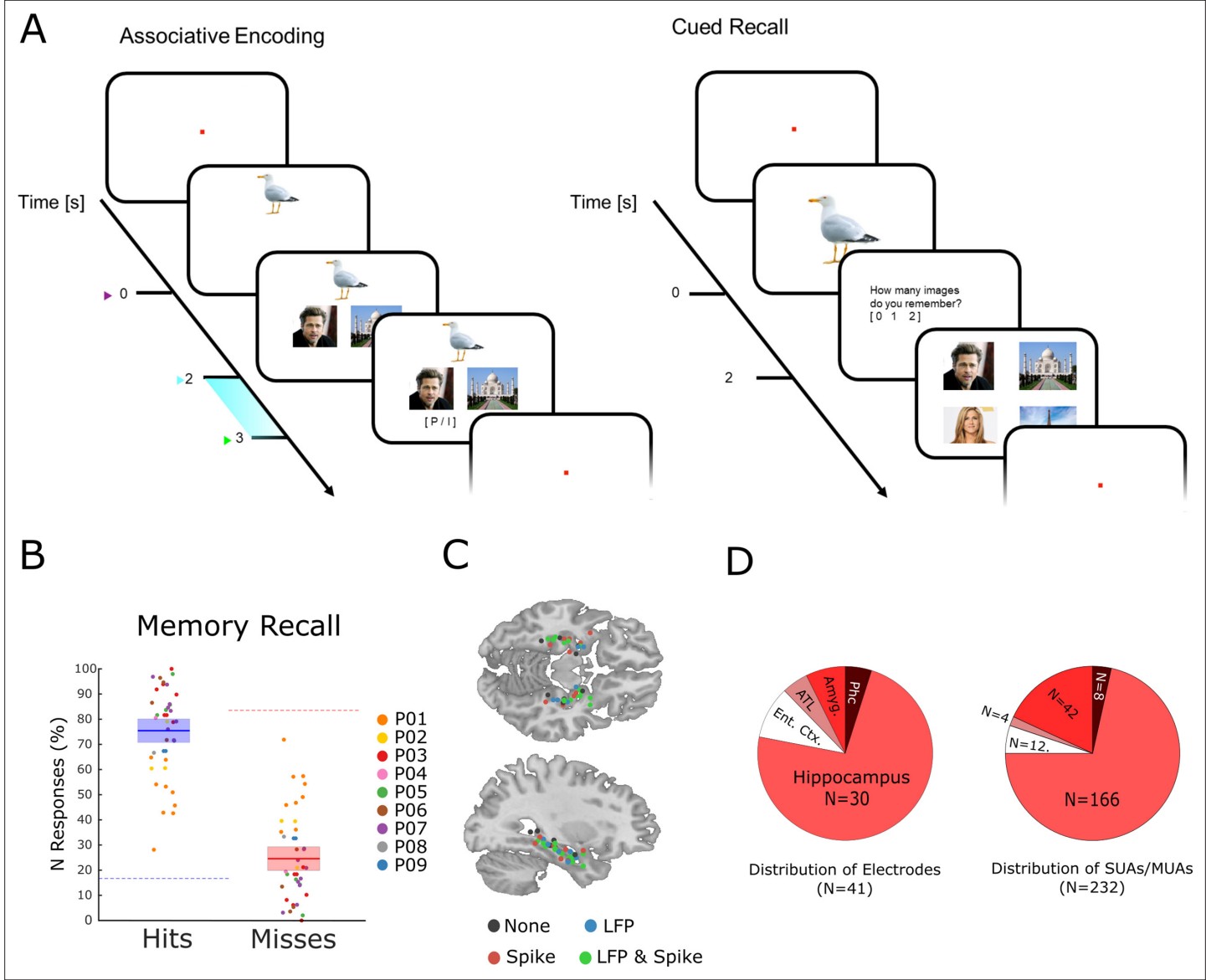

**Figure 1.** The memory task and behavioral results. (**A**) During encoding patients had to memorize three associated stimuli consisting of an animal, and either a pair of face images, a pair of place images, or a pair of face-place images. The light blue bar highlights the time window that was used for the analysis of LFP and neural spiking data. (**B**) Memory performance during the cued recall test is shown for all patients and sessions. Note that chance level in this task for hits and misses is 16.6 and 83.3%, respectively (indicated by dashed horizontal lines), and not 50% for both (see Methods for further details). (**C**) Electrode locations are plotted overlaid onto a template brain in MNI (Montreal Neurological Institute) space. Color codes indicate whether an electrode provided LFP, spiking, both, or no data. (**D**) Distribution of electrodes and recorded single- and multi-units is shown across medial temporal lobe regions (Ent. Ctx.: entorhinal cortex; ATL.: anterior temporal lobe; Amyg.: amygdala; Phc.: parahippocampal cortex; SUA: single unit activity; MUA: multi unit activity).

The online version of this article includes the following figure supplement(s) for figure 1:

**Figure supplement 1.** Automatic classification of single- and multi-units according to *Tankus et al., 2009*.

**Figure supplement 2.** Firing rate effects during memory encoding.

**Figure supplement 3.** Stimulus-evoked LFP activity is shown by means of inter-trial phase coherence (ITPC) and event-related potentials (ERPs).

**Figure supplement 4.** LFP power and inter-trial phase coherence (ITPC) results are shown.

co-firing of neurons consistent with synaptic plasticity principles such as spike-timing-dependent plasticity (STDP; *Bi and Poo, 2001*).

## Results

### Memory task and behavior

Nine patients with refractory epilepsy participated in 40 sessions of an associative episodic memory task (*Figure 1A*). During the encoding phase of the task, the patient was presented with several trials each containing a picture of an animal (cue), which was shown for 2 s. Then a pair of images appeared which either showed a face and a place, two faces, or two places. The patients were instructed to link the three elements of the episode together by mentally imagining a narrative (e.g. 'I saw a tiger in the zoo with Stephen Fry') and press a button to indicate whether the invented narrative or combination of images was plausible or implausible, then the next trial followed. All images were trial unique. After the encoding phase and a brief distractor test, memory performance was assessed by means of a cued recall test. During the test phase, the picture of the animal was presented for 2 s, and the patient indicated how many stimuli they could remember (0, 1, or 2). If they indicated to remember at least one image then a screen with four images appeared, and the patient selected the two images that they thought were paired with the cue originally. Trials for which both images were correctly recalled are labeled 'hit', all other trials (i.e. one image or both wrong) are labeled 'miss'. Therefore, contrasting hits with misses isolates neural processes which support the complete memorization of an episode (as opposed to incomplete memories or no memory at all). Any such process has to start when enough information is available for the patient to imagine the episode, which is at the onset of the paired images (i.e. 2 s after the onset of the cue). Therefore, all subsequent analysis focused on the time window that followed the onset of the paired images (2–3 s; highlighted in *Figure 1A*).

On average, patients correctly recalled both associated items on 75.43% (s.d.: 13.3) of the trials (*Figure 1B*). Note that this is well above chance level (16.6%). The remaining miss trials were approximately evenly distributed between incomplete memories (i.e. only one association recalled; 12.6%) or completely forgotten (i.e. both incorrect; 11.9%). Across sessions the proportion of full misses (i.e. both incorrect) was significantly below chance ($t_{39}=-1.92$; $p<0.05$). However, the proportion of fully forgotten trials appears to be higher than expected purely by chance. This is likely driven by a tendency of participants to either fully remember an episode, or completely forget it, as demonstrated previously in behavioral work (*Joensen et al., 2020*). Investigation of differences between the two types of misses (i.e. both incorrect versus one incorrect) was not feasible due to insufficient trial numbers.

### Within-region spike-LFP coupling to fast gamma oscillations correlates with memory

Neural spiking and LFP activity were recorded with Behnke-Fried hybrid depth-electrodes from MTL regions (*Figure 1C*). Most electrodes (73%) were located in the hippocampus, the rest was located in adjacent MTL regions. Altogether 232 putative single and multi-units were recorded of which 218 were used for further analysis (14 units were rejected because of too low firing rates; see Methods). These units were classified into single-units and multi-units using an automatic procedure based on waveshape homogeneity and inter-spike intervals (ISIs; *Tankus et al., 2009*), resulting in 82 putative single-units and 136 putative multi-units (*Figure 1—figure supplement 1*). Neural firing during encoding was not modulated by memory for the time window of interest (2–3 s). However, hits showed a sustained increase in firing rate compared to misses at a later time window (>3 s; *Figure 1—figure supplement 2*). LFPs for hits and misses also did not differ in terms of event-related potentials (ERPs) or inter-trial phase coherence (*Figure 1—figure supplements 3–4*). However, expected differences between hits and misses were obtained in broad band power (*Burke et al., 2015*), with hits showing decreased low frequency but increased high-frequency power.

Synchronization of neural firing of a single- or multi-unit to the population activity can be measured with spike-field coupling (SFC). SFC can occur at two different spatial levels, locally (within a region) or distally (across regions; *Figure 2A*). Locally, SFC indicates the firing of a neuron (or neurons for multi-units) being entrained to its surrounding LFP. Distally, SFC likely indicates that the firing of a neuron (or neurons) elicits post-synaptic currents in another region. Because postsynaptic currents

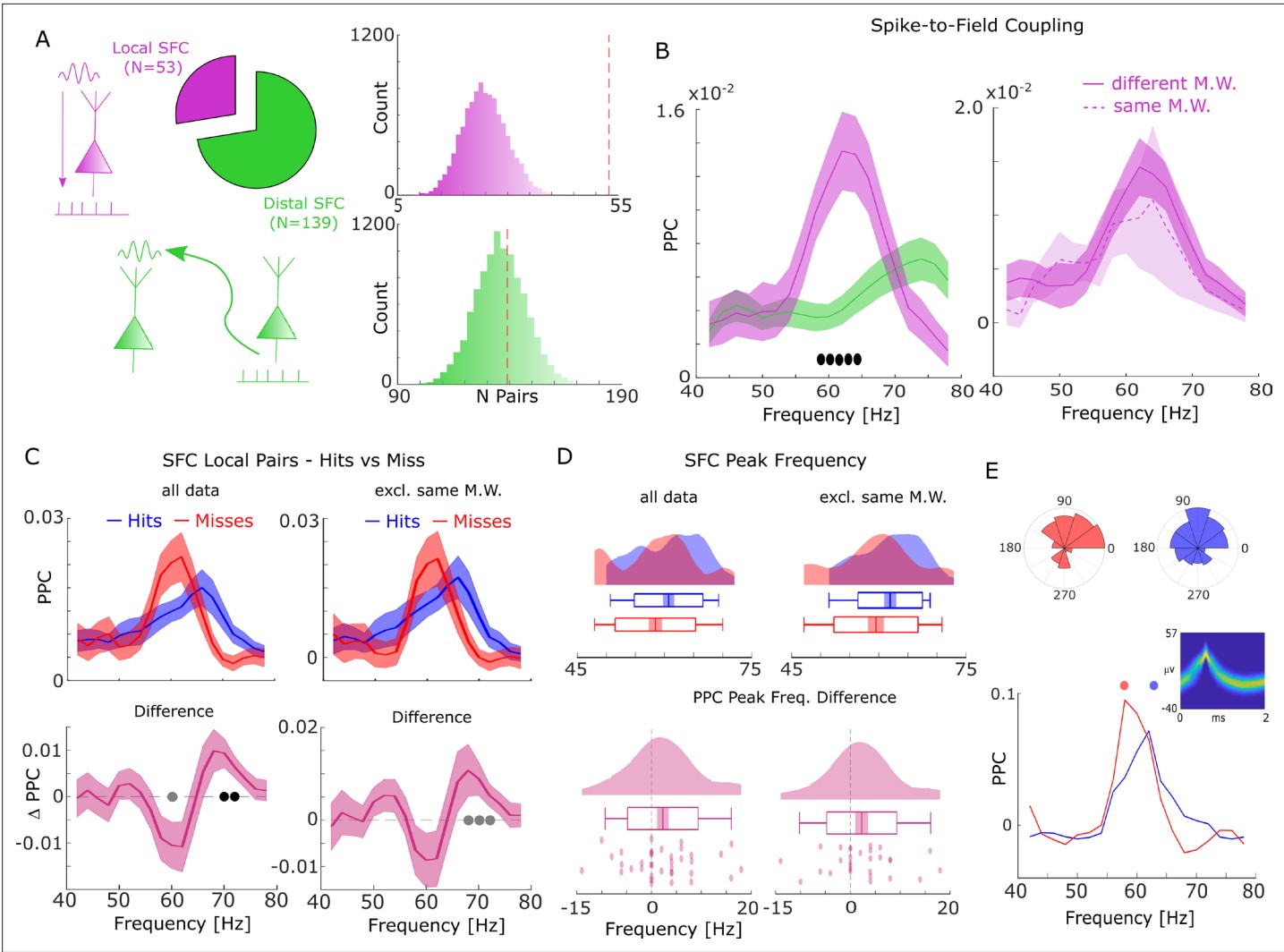

**Figure 2.** Spike-field coupling (SFC) results for gamma. (**A**) Number of significant ($p_{corr}$ <0.05) locally (pink) and distally (green) coupled spike-field pairs are shown. The histograms on the right show the results of a randomization procedure testing how many pairs would be expected under the null hypothesis. (**B, left**) Pairwise phase consistency (PPC) is plotted for local and distal spike-field pairs. Filled circles indicate significant differences ($p_{corr}$ <0.05). Shaded areas indicate standard error of the mean. (**B, right**) PPC is shown for channels where the spike and LFP providing microwire were the same (dashed line), or where they came from different microwires (solid line). (**C**) PPC is shown separately for hits and misses (top panel), and for the difference between the two conditions (bottom panel) for locally coupled spike-field pairs. The data from all local spike-LFP pairs is shown on the left, whereas on the right data where spikes and LFPs come from the same microwire were excluded. Filled black circles indicate significant differences ($p_{corr}$ <0.05). Gray circles indicate statistical trends ($p_{uncorr}$ <0.05). Shaded areas indicate standard error of the mean. (**D**) Peak frequency in PPC across all spike-field pairs is shown for hits and misses (top), and for the difference (hits-misses, bottom). The data from all local spike-LFP pairs is shown on the left, on the right data where spikes and LFPs come from the same microwire were excluded. The solid bar indicates the mean, shaded areas indicate standard error, the box indicates standard deviation, and the bars indicate 5th and 95th percentiles. (**E**) Local gamma SFC is shown for one example multi-unit recorded from the entorhinal cortex. Phase histograms on top indicate phase distribution for hits at 62 Hz (blue) and misses at 58 Hz (red). Spike wave shapes on the right are plotted by means of a two-dimensional histogram.

The online version of this article includes the following figure supplement(s) for figure 2:

**Figure supplement 1.** Selection bias control analysis results of a control analysis are shown to rule out a possible bias on the spike-field coupling results due to unbalanced trial numbers.

**Figure supplement 2.** Simulation of the effects of a non-stationary oscillator on wavelet analysis.

**Figure supplement 3.** Spike-LFP coupling results obtained with bandpass filtering and Hilbert transformation.

**Figure supplement 4.** Power for phase providing gamma frequencies.

**Figure supplement 5.** Relationship between gamma spike-field coupling (SFC) and power.

**Figure supplement 6.** Further control analyses for possible spike-interpolation artifacts.

*Figure 2 continued on next page*

*Figure 2 continued*

**Figure supplement 7.** Local gamma spike–LFP coupling results split by anatomical regions.

reflect the aggregated input to a neuron, distal SFC is usually interpreted as a functional measure of connectivity, with the spike providing region being the up-stream sender and the LFP providing region being the down-stream receiver (*Buzsáki and Schomburg, 2015*; *Liebe et al., 2012*; *Jacob et al., 2018*). Accordingly, we split spike-LFP pairs into these two categories, i.e., local and distal couplings. SFC was measured with the pairwise phase consistency (PPC) index (*Vinck et al., 2010*), which is not biased by the number of observations (e.g. spikes and trials). For distal couplings all possible pairings were considered; that is no constraints were imposed based on anatomy. Connectivity was therefore measured in a purely data-driven way.

During the time window of interest (2–3 s), 192 significantly (Rayleigh test; $p_{corr}$ <0.05; FDR-correction) coupled spike-LFP pairs were found in the high-frequency range (40–80 Hz), of which 53 were coupled to the local LFP (4.87% of all possible combinations) and 139 coupled to distal LFPs (2.49% of all possible combinations, *Figure 2A*). The number of locally coupled pairs was significantly higher than chance (Randomization test; p<0.0001), whereas the number of distally coupled pairs was not (p>0.35). Local SFC showed a pronounced peak in the fast gamma range (~65 Hz), which was substantially stronger compared to distal couplings (T-test; $p_{corr}$ <0.05; *Figure 2B*; FDR-correction). When calculating local spike-LFP coupling it is necessary to interpolate spikes to prevent high frequency artifacts (*Jacobs et al., 2007*); however, this interpolation can again potentially introduce artifacts and inflate spike-LFP coupling, especially for channels where spikes are coupled to the LFP of that same channel (which were ~20% of the data). To address this issue we split the local spike-LFP coupling data into channel pairs where the spikes and LFPs were measured on the same microwire, and where spikes and LFPs were measured on different microwires (but on the same bundle of B-F electrodes; see *Figure 2B*, right). The PPC profile for both was highly similar, suggesting that spike interpolation did not artificially inflate the spike-LFP coupling. Importantly, the peak frequency of local SFC varied as a function of memory formation such that hits showed stronger SFC at a higher frequency (~70 Hz) than misses (~62 Hz; T-test; $p_{corr}$ <0.05; FDR-correction; *Figure 2C*). This effect was also significant when using sessions as random variable (T-test; $t_{19}$=2.21; p=0.02; Cohen's d=0.49). A similar yet slightly weaker effect was also observed when excluding data where spikes and LFPs were measured on the same microwire (see *Figure 2C*, right; T-test; $p_{uncorr}$ <0.05).

This pattern suggests a shift in frequency, with hits showing SFC at a higher gamma frequency compared to misses. To test whether this shift of gamma frequency occurred consistently across spike-LFP pairs, a peak detection analysis was conducted where gamma peak frequencies for hits and misses were extracted and compared for each pair. The results confirmed that hits were characterized by significantly faster gamma frequencies compared to misses (T-test; $t_{36}$=1.96; p=0.029; Cohen's d=0.32; *Figure 2D* - left). Again, excluding data where spikes and LFPs were measured on the same microwire replicated this shift in peak frequency for hits compared to misses (t28=1.75; p=0.045; Cohen's d=0.32; *Figure 2D* - right). *Figure 2E* shows this effect for one example multi-unit, which couples to a slightly slower gamma rhythm for misses compared to hits. A control analysis, which effectively controls for a possible selection bias due to unbalanced trial numbers revealed similar results (*Figure 2—figure supplement 1*).

A set of control analyses were carried out to address possible concerns about non-stationarity of the signal. First, we carried out a series of simulations to ensure that the wavelet filters used here yield correct phase estimates for non-stationary signals (*Figure 2—figure supplement 2*). Second, we repeated the analysis using a Hilbert transform instead of a wavelet transform in combination with a bandpass filter (band-width of 4 Hz for theta and 8 Hz for gamma; see *Figure 2—figure supplement 3*). We also analyzed the power spectra of the LFP signal at spike times to ensure the presence of a meaningful physiological signal in the phase providing LFP signal (*Figure 2—figure supplement 4*), albeit this is not strictly needed for obtaining meaningful spike-LFP coupling results (*Bush and Burgess, 2020*; *Eliav et al., 2018*). We also tested whether a similar shift in peak gamma frequency as observed for spike-LFP coupling is present in LFP power, and whether memory-related differences in peak gamma spike-LFP are correlated with differences in peak gamma power (*Figure 2—figure supplement 5*). Both analyses showed no effects, suggesting that the effects in spike-LFP coupling were not coupled to, or driven by similar changes in LFP power. In addition, we also repeated the

above spike-LFP analyses using only LFPs for 'silent' microwires, i.e., channels where no spikes have been detected (*Figure 2—figure supplement 6*). Finally, we tested whether the absolute amount of spike-LFP coupling differed between hits and misses by comparing peak PPC values between hits and misses. No significant differences in peak PPCs were observed for the raw PPC values (T-test; $t_{36}=-1.7098$; p=0.09) and the selection-bias controlled data (T-test; $t_{36}=1.135$; p=0.26).

In the above analysis, all MTL regions were pooled together to allow for sufficient statistical power. Results separated by anatomical region are reported in *Figure 2—figure supplement 7* for the interested reader. However, these results should be interpreted with caution because electrodes were not evenly distributed across regions, and patients making it difficult to disentangle whether any apparent differences are driven by actual anatomical differences, or idiosyncratic differences between patients.

Taken together, the phase of fast gamma oscillations temporally organizes spikes within a region. Later fully remembered episodes (hits) are distinguished from incomplete or forgotten episodes (misses) by the frequency to which spikes are coupled to; with fast gamma oscillations benefiting memory formation, and slow gamma oscillations being detrimental for memory formation. This effect is unlikely to be caused by differences in stimulus-evoked activity since neither ERPs nor firing rates showed a memory-related difference in the time window of interest (*Figure 1—figure supplements 2–4*).

## Cross-regional spike-LFP coupling to fast theta oscillations correlates with memory

The same SFC analysis as above was carried out for the low-frequency ranges (2–40 Hz). We identified 103 locally coupled (9.46%) and 387 distally coupled (6.93%) spike-field pairs (Rayleigh-test; $p_{corr}<0.05$; FDR-corrected; *Figure 3A*). For both local and distal couplings, the number of significant pairs exceeded chance level (randomization test; both p<0.0001). Local couplings showed a peak PPC at around 5 Hz (and another peak at 13 Hz), whereas distal couplings showed a peak at around 8–9 Hz (*Figure 3B*). Local SFCs were robustly stronger than distally coupled pairs in the beta frequency range (20–30 Hz).

For local spike-LFP coupling, no significant differences ($p_{uncorr}>0.05$) between hits and misses were observed for the low-frequency range. In contrast, distally coupled spike-field pairs showed stronger coupling for hits compared to misses in the fast theta frequency range (8–10 Hz) and higher SFC for misses compared to hits in the slower theta frequency range (5 Hz; *Figure 3C*, T-test; $p_{corr}<0.05$; FDR-corrected). The stronger SFC in the fast theta band for hits compared to misses was also found to be significant when using sessions (T-test; $t_{20}=3.25$; p=0.002; Cohen's d=0.71) as random variable. No significant differences between hits and misses were obtained for locally coupled spike-field pairs.

Like the memory-related difference in gamma peak frequency, a shift in peak frequency also drove the memory-related difference in distal theta SFC. This was confirmed by a peak detection analysis showing that hits exhibited a slightly faster peak in theta SFC compared to misses (T-test; $t_{206}=3.49$; p=$2.99*10^{-4}$; Cohen's d=0.24; *Figure 3D*). This effect is shown for one example single-unit which is distally coupled to a slow theta oscillation for misses and to a fast theta oscillation for hits (see *Figure 3—figure supplement 1* for a control analysis on selection bias). As for gamma, control analyses addressing concerns about non-stationarity of the signal and the existence of meaningful theta power in the phase providing LFP signal are shown in *Figure 2—figure supplements 2 and 3*. Like the effects in local gamma coupling, these effects are unlikely to be due to changes in stimulus-evoked activity (*Figure 1—figure supplements 2–4*), and/or due to theta rhythmicity in the spiking of neurons themselves (*Figure 3—figure supplement 4*). As for gamma, we also tested whether a similar shift in peak theta frequency is present in LFP power, and whether there is a correlation between the memory-related differences in peak theta spike-LFP and peak theta power (*Figure 3—figure supplement 5*). Both analyses showed no effects, suggesting that the effects in spike-LFP coupling were not coupled to, or driven by similar changes in LFP power. We also repeated the above analysis for spike-LFP pairs by only using 'silent' LFP channels, i.e., channels where no SUA/MUA activity was detected (see *Figure 3—figure supplement 6*) to address possible concerns about artifacts introduced by spike interpolation. Finally, we report the distal spike-LFP results separated by anatomical region in *Figure 3—figure supplement 7*, which did not reveal any apparent differences in the memory-related modulation of theta spike-LFP coupling between regions.

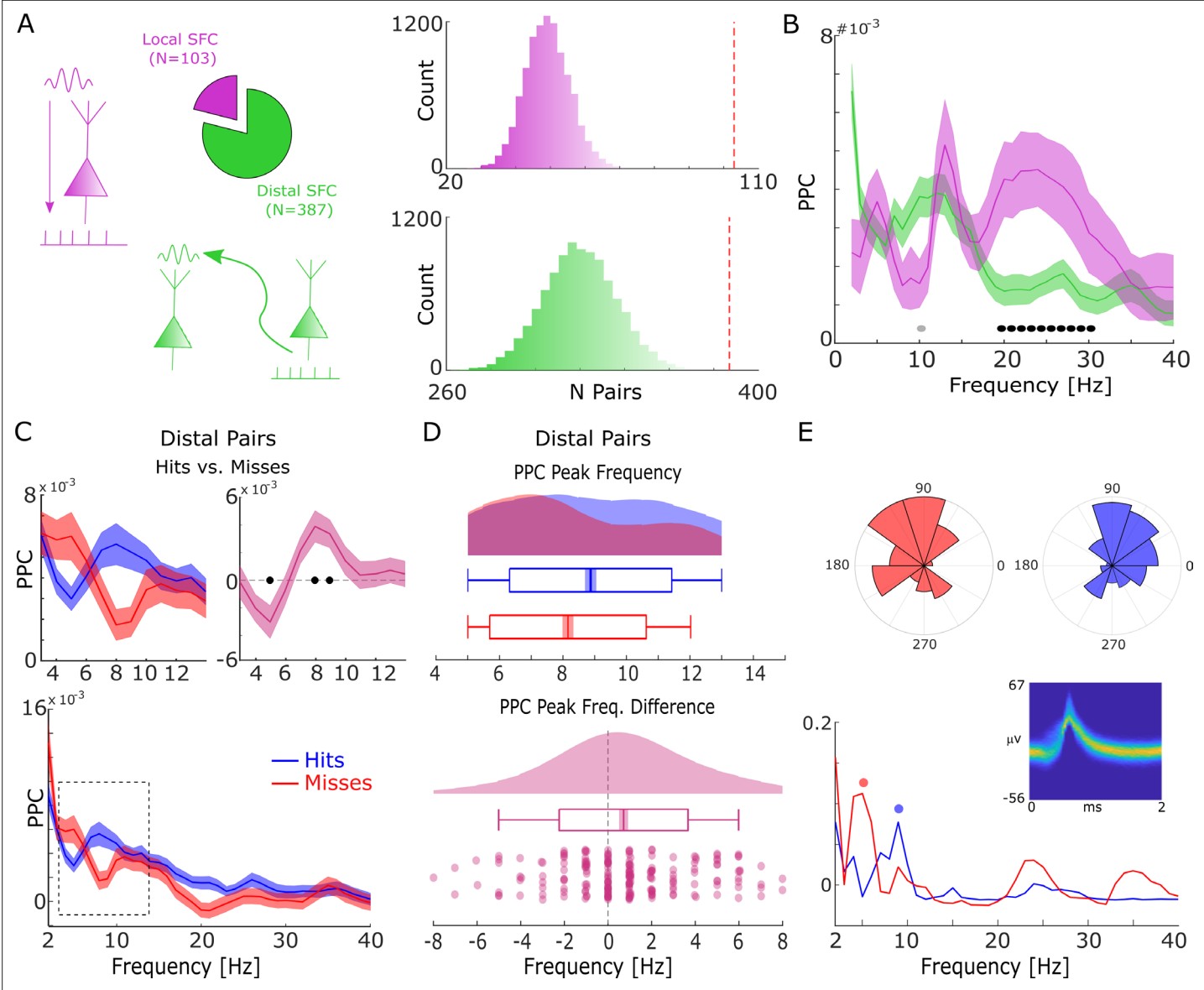

**Figure 3.** Spike-field coupling results for the lower frequencies. (**A**) A number of significant ($p_{corr}<0.05$) locally (pink) and distally (green) coupled spike-field pairs are shown. The histograms on the right show the results of a randomization procedure testing, with the red dashed line indicating the empirically observed value. (**B**) Pairwise phase consistency (PPC) is plotted for local and distal spike-field pairs. Black circles indicate significant differences ($p_{corr}<0.05$). Gray circles indicate trends ($p_{uncorr}<0.05$). Shaded areas indicate standard error of the mean. (**C**) PPC is shown separately for hits (blue) and misses (red), and for the difference between the two conditions (magenta) for distally coupled spike-field pairs. The top panels show PPC values for the theta frequency range, the bottom panel shows all frequencies up to 40 Hz. Shaded areas indicate standard error of the mean. Black circles indicate significant differences ($p_{corr}<0.05$). (**D**) Peak frequency in PPC across all distal spike-field pairs is shown for hits and misses (top), and for the difference (hits-misses). Box plots indicate the same indices as in *Figure 2D*. (**E**) Distal theta spike-field coupling is shown for one example single-unit recorded from the left posterior hippocampus, and the LFP recorded from the left entorhinal cortex. Phase histograms on top indicate phase distribution for hits at 9 Hz (blue) and misses at 5 Hz (red). Spike wave shapes on the right are plotted by means of a two-dimensional histogram.

The online version of this article includes the following figure supplement(s) for figure 3:

**Figure supplement 1.** Selection bias control analysis for distal theta spike-field coupling (SFC).

**Figure supplement 2.** Spike-LFP coupling results obtained with bandpass filtering and Hilbert transformation.

**Figure supplement 3.** Power for phase providing theta frequencies.

**Figure supplement 4.** Spike power analysis.

**Figure supplement 5.** Relationship between distal theta spike-field coupling (SFC) and power.

*Figure 3 continued on next page*

**Figure supplement 6.** Control analyses for possible spike-interpolation artifacts.

**Figure supplement 7.** Distal theta spike-LFP coupling results split by anatomical regions.

To assess the direction of information flow of distal theta spike-LFP coupling, and to test whether this flow of information has any bearing on memory performance, we compared the phase slope index (PSI) between hits and misses. Positive values indicate that the spike providing signal is the sender, and the LFP providing signal is the receiver and vice versa. The results indicate above chance PSIs have for hits peaking in the fast theta band (7–8 Hz). In addition, hits show significantly higher PSIs compared to misses (*Figure 4*), whereas PSIs for misses do not differ from zero. PSI results separated by anatomical regions are reported in *Figure 4—figure supplement 1*, which revealed that the PSI results were mostly driven by within regional coupling.

In agreement with the results obtained for local gamma oscillations we observed that distal theta spike-LFP coupling varied as a function of memory formation, with hits showing spike coupling at faster theta peak frequencies compared to misses. Thus, the present findings reveal two distinct cell populations that synchronize either to local gamma rhythms or distal theta rhythms, and a functional relationship between the peak frequency of gamma and theta phase coupling and memory formation. Importantly, directional coupling analyses demonstrate that the spike providing signal is the upstream sender, and the LFP providing signal is the downstream receiver. This result likely does not indicate that one single neuron can drive the entire LFP in a receiving region. Instead, it likely indicates that the spikes of a large population of neurons, of which we happen to sample one, provide input to a receiving region. This direction of information flow was dependent on memory success, suggesting that successful memory formation correlates with efficient cross-regional information transfer.

## Theta and gamma oscillations are coupled for hits but not for misses

The above results show that successful memory formation relies on gamma oscillations synchronizing neurons at a local level, and theta oscillations at ~8 Hz synchronizing neurons across regions. Intriguingly, peak frequencies of both oscillations showed a similar relationship with memory formation, with faster frequencies being associated with successful memory. This raises the question of whether gamma and theta oscillations are also temporally coordinated. For this analysis, we considered

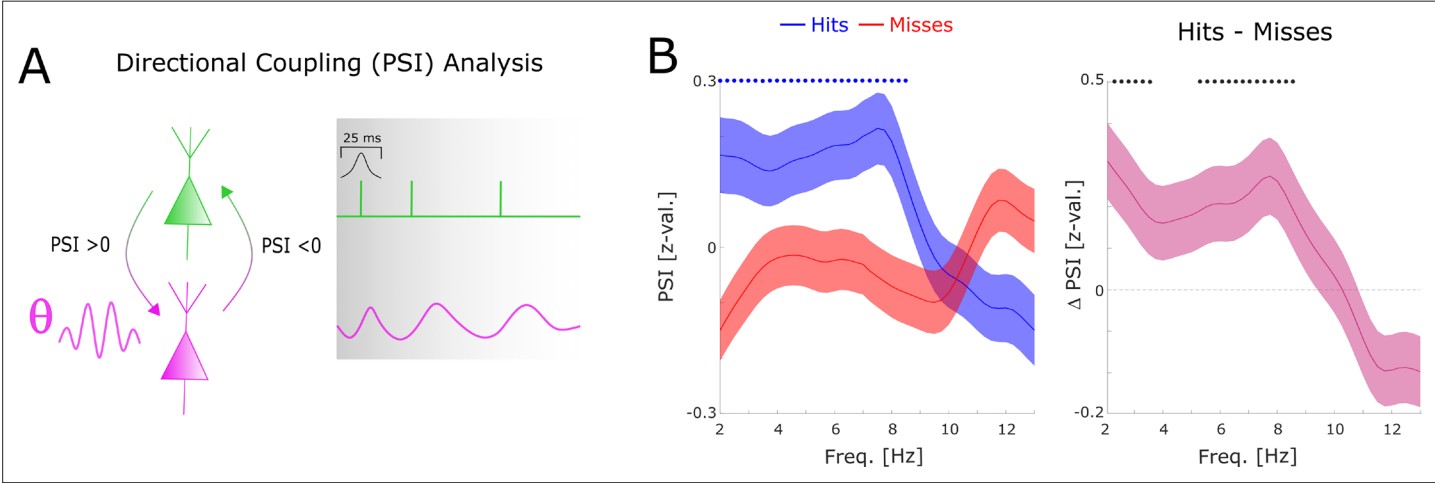

**Figure 4.** Directional coupling analysis of distal spike-LFP coupling using phase slope index (PSI). (**A**) A schematic of the analysis is shown. Spike time series were convolved with a Gaussian window, and the PSI was calculated between the spike time series (green) and the distally coupled LFP (pink). (**B**) The left plot shows normalized PSI (i.e. z-values) for hits (blue) and misses (red). Hits show significantly positive PSIs throughout the theta frequency range, peaking at ~8 Hz (blue circles; $p_{corr}$ <0.05). The right plot shows the difference in PSI between hits and misses. Hits show significantly higher PSIs compared to misses, especially in the high theta range (black circles; $p_{corr}$ <0.05).

The online version of this article includes the following figure supplement(s) for figure 4:

**Figure supplement 1.** Phase slope index (PSI) results split by anatomical regions.

**Figure supplement 2.** Phase slope index (PSI) results using a minimum of 30 spikes.

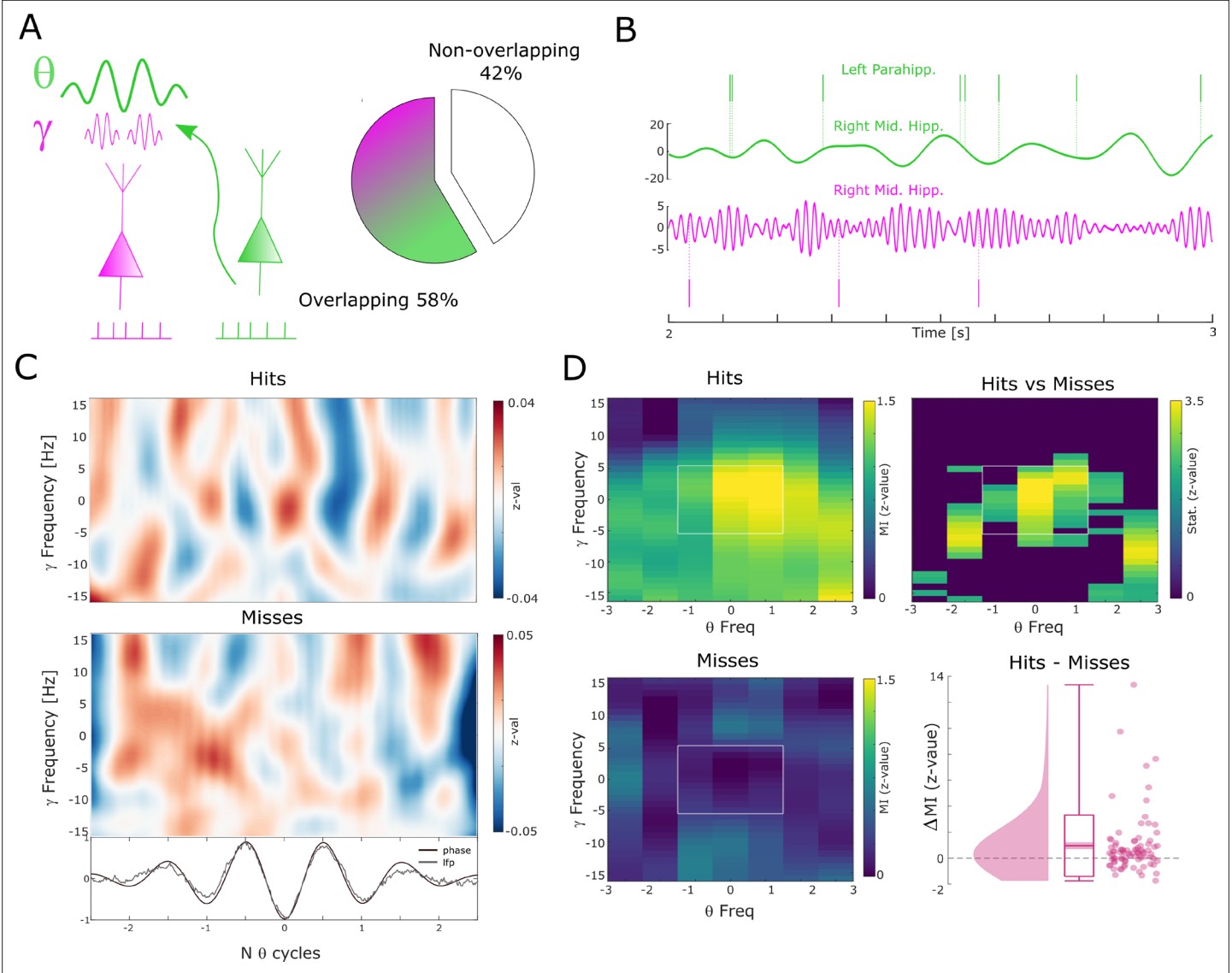

**Figure 5.** Theta to gamma cross-frequency coupling results. (**A**) Percentage of overlapping local gamma (pink) and distal theta (green) spike-field pairs is shown. (**B**) Spikes and band-pass filtered LFP data for one example single trial are shown. The top row shows spikes from a multi-unit in the left parahippocampal cortex which are coupled to the LFP in the right middle hippocampus (green). The gamma LFP from the same region (right mid hippocampus) is shown below (pink) as well as spikes from a multi-unit in the same region that is coupled to this gamma oscillation. Note the gamma power increase around theta troughs. (**C**) Theta phase sorted gamma power (y-axis centered to gamma peak frequency) is shown for all trials for the data shown in (**B**). The bottom panel shows averaged normalized band-pass filtered LFP data (black) and unfiltered LFP data (gray). (**D**) Co-modulograms are shown for hits and misses. Modulation indices (***Tort et al., 2010***), which indicate the strength of cross-frequency coupling, are plotted in terms of z-values where means and standard deviations were obtained from a trial shuffling procedure. The difference between hits and misses is shown as z-values obtained from a non-parametric Wilcoxon signrank test masked with $p_{corr} < 0.05$ (FDR-corrected). The panel in the bottom right shows the individual differences between hits and misses across the whole dataset (N=83 pairs).

The online version of this article includes the following figure supplement(s) for figure 5:

**Figure supplement 1.** Harmonic and asymmetric waveshape control analysis.

**Figure supplement 2.** Cross-frequency coupling results split by anatomical regions.

**Figure supplement 3.** Cross-frequency coupling results using only 'silent' microwire channels for gamma power estimation.

electrodes from regions where the LFP was locally coupled to spikes in the gamma range and distally coupled to spikes in the low-frequency (theta) range. More than half of the electrodes (58%) were available for this analysis (***Figure 5A***). CFC was calculated by means of phase-amplitude coupling using the modulation index (MI; ***Tort et al., 2010***). Importantly, theta and gamma frequencies were

adjusted to their peak frequency (see Methods) for each condition to account for the systematic difference in peak frequencies between hits and misses and to ensure the presence of a physiologically meaningful oscillation in both conditions (*Aru et al., 2015*) (see also *Figure 2—figure supplement 4* and *Figure 3—figure supplement 3*). Theta phase to gamma power coupling was evident in single trials (*Figure 5B-C*). Hits showed stronger theta phase to gamma amplitude coupling compared to misses (*Figure 5D*; Wilcoxon test; z=3.7; p=8.92*10$^{-5}$, Cohen's d=0.39). This increased CFC for hits compared to misses was also significant when pooling the data across sessions (Wilcoxon test; p=0.0386; Cohen's d=0.41). CFC can be subject to several confounds, which we addressed by a series of control analyses (see Methods and *Figure 5—figure supplement 1*). We also analyzed whether the memory-dependent effects of CFC differ between anatomical regions (see *Figure 5—figure supplement 2*). This analysis revealed that the results were mostly driven by the hippocampus; however, we urge caution in interpreting this effect due to the large sampling imbalance across regions. Finally, to address concerns about possible broadband power artifacts introduced by spike interpolation, we replicated the results by excluding high-frequency power providing channels with SUA/MUA activity (see *Figure 5—figure supplement 3*).

## Short co-firing latencies predict successful memory formation

One effect of the relative frequency increase of theta/gamma oscillations may be a more efficient transmission of information between neural assemblies. This can be demonstrated by considering two neurons which both have the same preferred phase of firing (e.g. maximal excitation), and which are both coupled by an oscillation with a constant phase lag of pi/4 (with neuron 1 leading and neuron 2 lagging). If the two neurons are coupled at a frequency of 8 Hz, then neuron 2 would fire ~31 ms after neuron 1 (i.e. 8 Hz equals a period length of 125 ms, divided by 4 = 31.25). If the neurons are, however, coupled at 4 Hz then neuron 2 would fire ~62 ms after neuron 1. To test this hypothesis, we analyzed neural co-firing at different time lags by computing the cross-correlation of spike trains between theta up-stream single-/multi-units (i.e. the distally coupled unit) and their corresponding gamma down-stream single-/multi-unit (i.e. the locally coupled unit; *Figure 6A*). Overall, 32 pairs were available for this analysis, 24 of which showed above threshold co-firing (see *Figure 6—source data 1* for detailed information about each pair). Cross-correlations for hits and misses were each compared to a trial-shuffled baseline and transformed to z-scores effectively eliminating biases introduced by different trial numbers.

Compared to baseline, hits showed significant above chance co-firing at lags 20–40 ms, whereas co-incidences for misses peaked at 60 ms (T-test; $p_{corr}$<0.05; FDR-correction; *Figure 6B*). In addition, hits showed stronger co-firing compared to misses at 20 ms (T-test; $p_{corr}$<0.05; FDR-correction; *Figure 6B*). A peak detection analysis revealed that co-firing for hits peaked significantly earlier compared to misses ($t_{21}$=−3.201; p=0.004292; Cohen's d=−0.68; *Figure 6D*). This result held also when using a more conservative approach, i.e., pooling the data across number of single-/multi-units ($t_{11}$=−3.337; p=0.00663; Cohen's d=−0.96). Intriguingly, this memory related co-firing effect was observed only when selecting pairs of single-/multi-units that were both locally coupled to gamma and distally coupled to theta. Analyzing all possible pairs of distally coupled theta units showed no differences in peak co-incidences between hits and misses ($t_{126}$=−0.78; p=0.432). In addition, a completely unconstrained co-firing analysis where all pairs possible pairings of units were considered also showed no systematic difference in co-firing lags between hits and misses (*Figure 6—figure supplement 1*). This is quite remarkable given that statistical power for these latter two analyses was substantially higher. This pattern of results suggests that the coupling of down-stream neurons to local fast gamma oscillations is crucial for observing the memory-dependent effect of co-firing at critical time windows. To test for a similar effect in the reverse direction (i.e. local gamma coupled unit --> distal theta coupled unit), the same co-incidence analysis was carried out for negative lags. Intriguingly, misses showed peak co-firings at shorter negative latencies (i.e. closer to 0) compared to hits (*Figure 6—figure supplement 2*; $t_{14}$=−2.866; p=0.0124; Cohen's d=−0.74). This result is consistent with the STDP framework, whereby a negative time lag leads to a decrease of synaptic connectivity (*Markram et al., 1997*; *Bi and Poo, 2001*). As for the above analysis, we also investigated any apparent differences in co-firing between anatomical regions. These results are reported in *Figure 6—figure supplement 3* and show that the earlier co-firing for hits compared to misses was approximately equivalent across regions. The co-firing analyses were replicated with different smoothing parameters (see *Figure 6—figure*

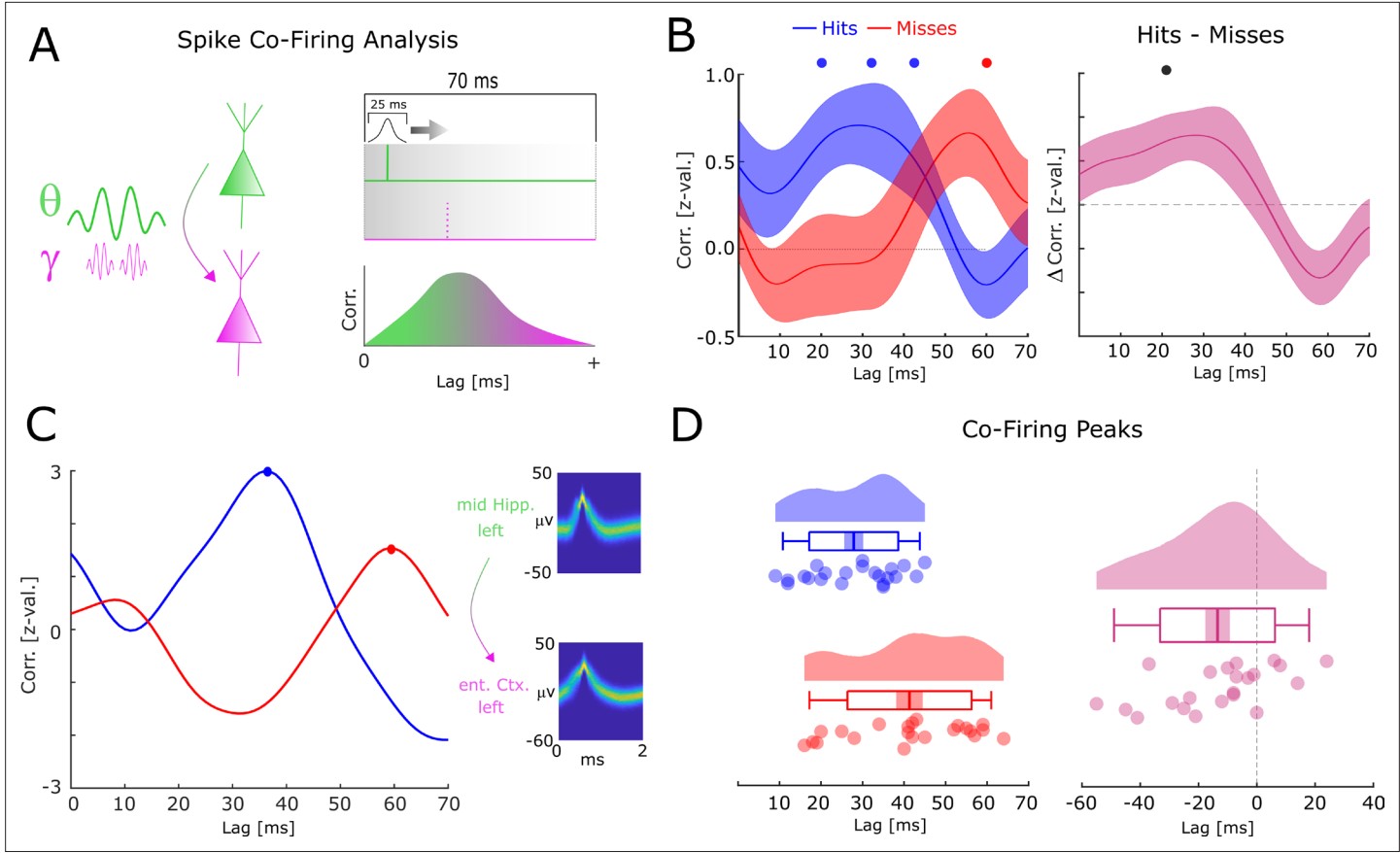

**Figure 6.** Co-firing analysis results for theta-gamma-coupled assemblies. (**A**) A schematic of the co-firing analysis is shown. Pairs of putative up-stream (green) and putative down-stream (pink) units were selected for the co-firing analysis. Co-firing was measured by cross-correlating spike time series (convolved with a Gaussian envelope). Cross-correlations indicate the latency of firing of a putative down-stream neuron (pink) in response to a putative up-stream neuron (green). (**B**) Spike cross-correlations for hits and misses are plotted in terms of z-values derived from a trial shuffling procedure. Hits (blue) show increased co-firing between putative up-stream and putative down-stream neurons at around 20–40 ms ($p_{corr}$ <0.05), whereas misses (red) peak at 60 ms ($p_{corr}$ <0.05). Shaded areas indicate standard error of the mean. Differences between co-firing of hits and misses are plotted on the right. Hits show higher co-firing at 20 ms compared to misses ($p_{corr}$ <0.05). (**C**) Co-firing data is shown for one example pair of units. (**D**) Results of the co-firing peak detection analysis. The distribution of the peak lag is shown for hits (blue) and misses (red), and for the difference for each pair of neurons (pink). Hits exhibit significantly shorter lags of co-firing compared to misses (p<0.005).

The online version of this article includes the following source data and figure supplement(s) for figure 6:

**Source data 1.** Table of regions where distally and locally coupled neurons were recorded.

**Figure supplement 1.** Co-firing analysis using all possible pairs of SUAs/MUAs.

**Figure supplement 2.** Co-firing analysis at negative lags.

**Figure supplement 3.** Co-firing analysis split by anatomical regions.

**Figure supplement 4.** Co-firing analysis for different lengths of gaussian windows (i.e. smoothing).

*supplement 4*). No significant differences in firing rates between hits and misses were found (p>0.3), and on correlations between firing rates and the co-firing latencies were obtained (R=−0.06; p>0.7), suggesting that firing rates had no influence on the observed co-firing differences between hits and misses.

Together, these results suggest that successful memory formation correlates with shorter latencies of co-firing between down-stream and up-stream neurons. Notably, this effect is selective for pairs of single-/multi-units that are both distally theta-coupled and locally gamma-coupled.

## Discussion

We investigated neural firing, spike-LFP coupling, and co-firing of single-/multi-units during the early encoding stage of complex associations into episodic memory. Our results demonstrate that successful memory formation during this early stage is correlated with a fine-grained pattern of local and distal neural synchronization reflected predominantly in theta and gamma oscillations. On a local level, neural firing coupled to relatively fast gamma oscillations predicts memory success, whereas coupling to relatively slower gamma oscillations predicts memory errors. A similar relationship was obtained on a distal level (i.e. spikes and LFPs recorded from different electrodes) where coupling at relatively faster theta oscillations occurred during hits, and coupling at relatively slower theta oscillations was observed during misses. Directional connectivity analyses suggested that the single-/multi-units were the up-stream sender and the LFP the down-stream receiver. Furthermore, gamma oscillations were coupled to the phase of theta, particularly for hits but not for misses, suggesting that successful memory formation is related to the coordination between the two oscillations. Finally, and most crucially, we found that the co-firing of pairs of single-/multi-units was correlated with memory formation such that co-firing at short but not long latencies predicted successful memory. These findings link theta and gamma oscillatory dynamics to neural firing and are consistent with current theories that emphasize the role of oscillations for synchronizing neural assemblies to support memory processes (*Fell and Axmacher, 2011*; *Buzsáki, 2010*; *Jutras and Buffalo, 2010*; *Hanslmayr et al., 2016*).

Enhanced local spike-LFP coupling for later successfully recognized pictures in the high gamma but not low gamma band has been observed in the hippocampal formation of macaques (*Jutras et al., 2009*). We replicate this finding here in humans with an associative episodic memory task. However, in contrast to this previous study we observed a shift in the peak gamma frequency rather than just an increase in spike-LFP coupling which could be due to differences in the task requirements, recording location, analysis techniques, or species-specific differences. Other studies in rodents have identified different roles for fast vs slow gamma oscillations in the hippocampus, with fast gamma oscillations reflecting the routing of information into the hippocampus and slow gamma oscillations reflecting the routing of information out of the hippocampus (*Colgin et al., 2009*). Consistent results were obtained in humans showing that fast gamma oscillations positively predict encoding, whereas slow gamma oscillations positively predict retrieval of memories (*Griffiths et al., 2019*). Our results are consistent with these studies in suggesting that relatively fast gamma oscillations, but not slow gamma oscillations, are beneficial for memory encoding and demonstrate this phenomenon on the level of spike-LFP coupling. As suggested in theoretical papers (*Buzsáki, 2010*; *Jutras and Buffalo, 2010*), the coupling of spikes at fast gamma oscillations may support memory formation because of the precise temporal packaging that it entails. Such tighter temporal compression of spikes increases the chances of an up-stream neural assembly to drive their down-stream partners (*Buzsáki, 2010*). Indeed, this was observed in the co-firing analysis where we found shorter co-firing lags for successful memory trials compared to unsuccessful memory trials. To this end, our results are consistent with the idea that fast gamma oscillations correlate with more efficient neural communication, therefore supporting memory formation processes. However, it must be acknowledged that our results do not show a direct correlation between gamma oscillations and the lag of co-firing, let alone a causal role of fast gamma oscillations for short latency co-firing. Such relationships should be investigated in future studies.

Paralleling the results for local gamma spike-LFP coupling, a frequency shift was also observed for distal theta spike-LFP couplings. Here, coupling at relatively faster theta oscillations correlated with later memory success, whereas coupling at relatively slower theta oscillations correlated with memory errors. Dissociations between slow and fast theta oscillations in humans have been reported previously (*Lega et al., 2012*; *Goyal et al., 2020*), albeit slow theta in these studies was considerably slower than reported here (i.e. ~2–4 Hz as opposed to 4–6 Hz). Therefore, the relatively slower theta reported in the current experiment may not reflect this classic slow theta frequency band (*Jacobs, 2014*). Instead, they may reflect frequency differences of spike-LFP coupling within the faster theta (4–10 Hz) band which may change as a function of cognitive states, such as memory outcome. It is well documented that the frequency of theta correlates positively with running speed in rodents (*Sławińska and Kasicki, 1998*) and humans (*Goyal et al., 2020*), and that running speed-induced changes correlate with memory outcome (*Richard et al., 2013*). A recent theoretical paper speculated that variations in running speed in rodents may reflect different levels in excitatory input to the hippocampal system,

which in humans can be equated with attention (*Buzsáki and Tingley, 2018*). It is therefore conceivable and consistent with computational modeling studies (*Lefebvre et al., 2015*), that the frequency of theta is subject to modulation of the level of excitation or cognitive states which correlate with memory outcome. On a functional level, an increase of frequency in theta spike-LFP coupling could have the same role on neural signal transmission as observed for gamma, i.e., enhancing the likelihood of an up-stream sender to drive their down-stream partners by temporal compression of spikes. Consistent with this interpretation, increased directional coupling from spikes to LFPs was observed for theta for successful memory trials (*Figure 4*). This result is in line with previous studies demonstrating a crucial role of inter-areal theta phase synchronization for memory formation (*Solomon et al., 2019*). However, no attempt was made here to analyze long-range connectivity in a fine-grained anatomical manner, i.e., by differentiating between hippocampal subfields or layers due to difficulties in asserting the exact locations of microwires. Instead, connectivity patterns between spike-providing and LFP-providing electrodes were analyzed in a purely data-driven way. This is a limitation that should be addressed by future studies using sophisticated localization techniques.

Theta and gamma oscillations not only showed parallel relationships between spike-LFP frequency and memory but also demonstrated cross-frequency phase-to-amplitude coupling. Gamma amplitude co-fluctuated with theta phase with the degree of this CFC being positively related to memory (i.e. stronger theta-gamma coupling for hits compared to misses). This result is in line with a previous study reporting similar results (*Lega et al., 2016*) and extends these findings to microwire recordings and links it with spike-LFP coupling. Furthermore, this result shows that the amplitude of local gamma oscillations rhythmically synchronizes to the phase of theta which mediates neural synchrony between regions, particularly if memory formation is successful. This result is consistent with current network models where fast oscillations regulate local connectivity, slower oscillations regulate long-range connectivity, and CFC enables efficient interfacing between the local gamma and the long-range theta networks (*Lisman and Jensen, 2013*; *Roux and Uhlhaas, 2014*; *von Stein and Sarnthein, 2000*).

The net result of a tighter synchronization of spikes locally at gamma, distally at theta, and increased synchronization between theta and gamma, is likely to be increased efficiency in neural communication. One way to quantify the efficiency of neural communication is to measure the time it takes for an up-stream neural assembly to drive their down-stream partners. Indeed, we observed that co-firing between neurons occurred at earlier lags for successful memory trials compared to erroneous memory trials. On a synaptic level, such co-firing at earlier lags would lead to strengthening of synaptic connections with the amount of strengthening decaying exponentially with the lag – known as STDP (*Bi and Poo, 2001*). These results are consistent with previous findings (*Dickey et al., 2021*) and support the idea that oscillations play a crucial role for memory formation because they enable efficient signal transmission and thereby affect synaptic plasticity (*Fell and Axmacher, 2011*; *Buzsáki, 2010*; *Jutras and Buffalo, 2010*).

The idea that synchronized inter-regional oscillations reflect effective communication has recently been questioned by a study showing that inter-regional phase synchronization can be a consequence rather than the cause of connectivity (*Schneider et al., 2021*). Oscillatory activity in a local circuit will be reflected in postsynaptic activity (i.e. the LFP) of any area that it projects to. Consequently, giving rise to phase locking in the LFP between the two areas which highlights a weakness of LFP-based connectivity measures and raises the need for additional methods to disambiguate between scenarios where oscillations establish communication between regions, and where they simply are a consequence thereof. To this end, we not only report a memory-dependent shift from slower to faster frequencies in theta and gamma spike-LFP coupling but critically also report a memory-dependent shift in spike-to-spike coupling, with hits showing earlier co-firing compared to misses. This finding is consistent with the idea that nested coupling of fast theta and gamma oscillations enables efficient neural communication. However, whether this shift of co-firing lag is caused by a speed up of theta and gamma oscillations remains an open question.

Notably, the neural co-firing analysis indicates a bidirectional flow of information between the hippocampus and surrounding MTL areas, such as the entorhinal cortex (see *Figure 6—figure supplement 3*; *Figure 6—source data 1*). This result parallels other studies in humans showing that successful encoding of memories depends not only on the input from surrounding MTL areas into the hippocampus but also on the output of the hippocampal system into those areas, and indeed on

the dynamic recurrent interaction between these input and output paths (*Maass et al., 2014*; *Koster et al., 2018*).

Our study also has several limitations and caveats that need to be considered when interpreting the results. First, sample size of patients is relatively low (N=9), however, not unprecedented in the field; a seminal study showing increased theta phase coupling of neural firing for later successfully remembered items reported 9 patients and 14 sessions (*Rutishauser et al., 2010*). To offset the low N in patients, several sessions were carried out per patient yielding 40 sessions overall. Second, single-units and multi-units were analyzed together. This was necessary because of the limited number of recorded single-units. Therefore, we cannot disambiguate between synchronization of neural populations or individual neurons, which is particularly relevant for the co-firing analysis. As a result, we cannot infer whether the observed oscillations co-occur with synchronized firing of pairs of individual neurons or synchronized firing between cell assemblies. Third, all effects reported here were observed in a relatively early time window, i.e., the first second after the stimuli appeared on the screen (2–3 s after cue-onset; see *Figure 1A*). This analysis window was chosen because it reflects the earliest possible time point when memory formation can happen, i.e., when the full information of the memory is presented. This might indicate that the observed differences between correct and erroneous memory trials reflect initial memory processing steps such as the routing of information into the appropriate MTL regions. Interestingly, these early modulations of neural synchronization by memory encoding were observed in the absence of modulations of firing rates, which is consistent with previous results in humans (*Rutishauser et al., 2010*) and macaques (*Jutras and Buffalo, 2010*), but contrasts with *Zheng et al., 2022*. Studies in macaques showed that attention increases spike-LFP coupling while not affecting firing rates (*Fries et al., 2001*). It is therefore conceivable that these initial network dynamics reflect attentional processes, which act as a gate keeper to the hippocampus and thereby set the stage for later memory forming processes (*Moscovitch, 2008*). However, the observed synchronization processes may be less reflective of the actual binding process per se, linking the stimuli into a coherent memory trace which supposedly happens at later processing stages (*Griffiths et al., 2021*). Further experiments are needed to address these issues in detail.

## Materials and methods
### Patients
Nine patients with refractory epilepsy volunteered to participate in the experiments. Mean age of patients was 37 years (s.d.: 9.1; range: 26–53). All but two patients were right-handed. On average, patients suffered 15.6 years from epilepsy. All patients had temporal lobe epilepsy with either a left (N=3), right (N=3)m or bilateral focus (N=3). Each patient participated in at least one experimental session, and in most cases, more than one session such that overall data from 40 sessions was available for analysis.

Patients were treated in one of three hospitals, the Queen Elizabeth University Hospital Birmingham (N=6), the Epilepsy Centre at the University Hospital in Erlangen (N=1), or the Vrije Universiteit Medisch Centrum Amsterdam (N=2). Ethical approvals were given by National Research Ethics Service (NRES), Research Ethics Committee (Nr. 15/WM/0219), the ethical review board of the Friedrich-Alexander Universität Erlangen-Nürnberg (Nr. 124_12 B), and the Medical Ethical Review board of the Vrije Universiteit Medisch Centrum (Nr. NL55554.029.15), for Birmingham, Erlangen, and Amsterdam, respectively. Informed consent to participate in the experiments and consent to publish the results was obtained from the patients prior to data collection.

### Task and procedure
All participants completed at least one session of an associative episodic memory task which required patients to form trial unique associations between three images (*Figure 1A*). In one session, several blocks of the memory task were carried out. One block comprised three phases, an encoding phase, a distractor phase, and a recall phase. During encoding, participants were first presented with an image cue of an animal for 2 s, followed by a pair of two images made up of any combination of a well-known face or a well-known place (i.e. face–place, face–face, or place–place pairs; presented for 2 s). The initial number of trials was set according to the patient's cognitive abilities as estimated by the experimenter. This number was then reduced if the hit rate fell below 66.25%, or increased if the hit

rate surpassed 73.75%, effectively adjusting task difficulty according to the participant's ability. On average, participants completed 19.1 trials (S.D.: 10.3; range: 2-82) per block. Participants were asked to vividly associate these triplets of images. Participants were encouraged to make up a story, which would link the three images to help them memorize the associations. For each triplet, participants were asked whether the story they came up with (or combination of pictures) was plausible or implausible. This plausibility judgment was used to keep participants on task rather than to yield a meaningful metric. Participants were self-paced in providing a judgment, and the following trial began immediately afterward. After encoding, the distractor phase was carried out which required participants to make odd/even judgments for 15 sequentially presented random integers, ranging from 1 to 99. Feedback was given after every trial. After completion of the distractor task, the retrieval phase commenced. Participants were presented with every animal image cue that was presented in the earlier encoding stage and, 2 s later, were asked how many of the associated face or place images they remembered (participants had the option of responding with 0, 1, or 2). If the participant remembered at least one image, they were then asked to select the pair of images from a panel of four images shown during the previous encoding block (two targets and two foils). The experimental script did not log how many images the patient indicated that they thought to remember. Foils were drawn from images, which were also presented in the preceding encoding phase but were paired with a different animal cue. Image positions on the screen were randomized for each trial. Therefore, a given association could either be remembered completely (i.e. both images correctly identified), remembered partially (i.e. only one image correctly identified), or fully forgotten (i.e. no image recalled or both incorrect).

Notably, the chance level in this task is 16.6%, and not 50% as one might initially assume. This is because the participant selects two stimuli out of four in two sequential steps. The chance of getting the first stimulus correct is 50% (i.e. 2 out of 4). The chance of getting the second stimulus also correct is 33% (i.e. 1 out of 3). The combined probability for both choices being correct therefore is 0.5 × 0.3 = 0.166 or 16.67%. Similarly, the chance of getting both stimuli incorrect is also 16.67%. Getting one stimulus correct is the most probable outcome with a likelihood of 66.67%. Therefore, partial hits/ misses (i.e. 1 correct, 1 incorrect) are likely to also contain a high proportion of lucky guesses. Partial hits/misses and full misses were combined into one miss category. Participants were self-paced during the recall stage, though the experiment ended after a runtime of 40 min in total. All participants completed the task on a laptop brought to their bedside.

## Recording data

To record behavioral responses and to present instructions and stimuli, a Toshiba Tecra laptop (15.6 inch screen) was used in the hospitals in Birmingham and Erlangen. In Amsterdam, an ASUS laptop was used (15.6 inch screen). All laptops operated on Windows 7, 64-bit. Psychophysics Toolbox Version 3 (*Brainard, 1997*) was used with MATLAB 2014 or MATLAB 2009b (Amsterdam). For responses, the following buttons were used: up-down-left-right arrows to select the images during the recall phase, and the 'End' key on the Numpad was used to confirm the selection. During encoding, the 'up' and 'down' arrow keys were used to give the plausible/implausible ratings, respectively. During the distractor phase, the 'left' and 'right' arrow keys were used to give the odd/even judgements, respectively.

Electrophysiological data were recorded from Behnke-Fried hybrid micro-macro electrodes (Ad-Tech Medical Instrument Corporation, Oak Creek, WI). Each Behnke-Fried hybrid electrode contained eight platinum-iridium high-impedance microwires with a diameter of 38 µm, and one low-impedance microwire with the same diameter extending from the tip. Different referencing schemes were used across the hospitals/patients in order to yield the best signal-to-noise ratio in the different environments (i.e. yield highest number of visible spikes in raw data). For five patients, the high-impedance contacts were referenced against either a low-impedance microwire or macro-contact, thus yielding a more global signal. For the remaining four patients, high-impedance contacts were referenced against another 'silent' (i.e. containing no visually detectable spikes) high-impedance wire, thus yielding a more local signal. As part of the pre-processing, data were re-referenced to yield a comparable local referencing scheme across all datasets (see below). The data were recorded continuously throughout the experiment on an ATLAS Neurophysiology system (Neuralynx Inc) with a sampling rate of 32 kHz (Birmingham and Amsterdam) or 32.768 kHz (Erlangen) and stored as a raw signal for processing and analysis.

## Data analysis

The code used for data analysis is available at https://osf.io/fngz8/.

### Behavior

Neural activity during the encoding phase was separated into hits and misses according to memory performance in the subsequent recall phase. Hits constitute trials where both items were later correctly retrieved (i.e. complete memory); misses constitute trials where at least one item was remembered incorrectly or where the patient indicated that they did not remember any item (i.e. incomplete memories or fully forgotten). Reaction times of the plausibility ratings indicate the time from onset of the cue (i.e. animal image) to the button press and were calculated per subject by using the median across trials. Reaction times were 15.01 s on average (s.d. 5.79) for hits and 15.26 s misses (s.d. 6.85) and did not differ significantly ($t_8$=0.44; p>0.5).

### Electrode localization

Electrodes were localized for using one of the following procedures. For patients recorded in Birmingham and Erlangen, pre- and post-implantation T1 structural MRIs (MP-Rage) were co-registered and normalized to MNI space using SPM8 (https://www.fil.ion.ucl.ac.uk/spm/software/spm8/). If both scans were available, the normalization parameters were estimated on the pre-implantation MRI, and these parameters were then applied on the post-implantation MRI. For two patients, only the post implantation MRI was available. For these two patients, normalization parameters were estimated and applied to this post-implantation MRI. The location of the microwire bundle was either clearly visible as an image artifact in the post-implantation MRI. If not, then the location of the microwire bundle was inferred visually by extrapolating the electrode trajectory by 5 mm.

For patients recorded in Amsterdam, pre-implantation structural T1 scans and post-implantation CT scans were available. These were overlaid and normalized using the same procedure as described above using SPM8. The microwire contacts were clearly visible in the CT scans, therefore the location could be estimated directly in all cases.

### Spike sorting

Spikes of recorded neurons were extracted offline from high-frequency activity (500 Hz–5 kHz). Spike detection and sorting were done using Wave_clus (*Quiroga et al., 2004*). All units with at least 50 spikes in either condition (hits, misses), a mean spike count >2, and mean firing rate >1 Hz during the encoding period (0–4 s) were submitted to further analysis. The resulting units were further visually inspected to reject noise based on waveshape, spike distribution across trials, and ISIs. Across all patients and sessions, 218 units were retained for analysis. As a final step, units were classified into single-units and multi-units following an automatic procedure developed by *Tankus et al., 2009*, which have been shown to closely match classification of trained researchers. This algorithm uses two criteria to classify units, which are ISIs and variability of the spike waveshape. Concerning the first criterion, a given unit is classified as a multi-unit if more than 1% of ISIs are smaller than 3 ms (which would violate the refractory period of firing of neurons). For the second criterion, the variability of the spike waveshape is computed in the time window of the rising flank of the spike waveshape (see Figure S1B). The end of the rise time was the peak of the spike waveshape. The beginning of the rise time was estimated following the procedure described in *Tankus et al., 2009* using the time-point of the maximum curvature (i.e. second derivative) within an area at the start of the spike wave-shape. Spike waveshape variability (i.e. criterion 2) was then calculated by dividing the sum of the standard deviation by the height of the spike wave.

$$c2 = \frac{\sum_i^j s(t_i)}{m(t_j) - m(t_i)}$$

where i = start of the rise time and j = end of the rise time, and m = voltage at time t. This second criterion, c2, can be understood as the inverse of the signal-to-noise ratio where a low value means low variability in spike waveshape. Following *Tankus et al., 2009*, we labeled a given unit as an SU if

c2 was <3. The distribution of both criteria across all units is shown in Figure S1A; waveshapes and ISIs for representative SUs and MUs are shown in Figure S1C.

## Firing rate and spike density

Time stamps of spikes during the encoding phase were extracted and converted to continuous time series containing 0 s (no spike) and 1 s (spike) at a sampling rate of 1 kHz. These time series were cut into trials with a duration of 14 s centered to cue onset (i.e. animal image), starting at −7 s. These trial-based spike time series were then convolved with a gaussian window of 250 ms length to yield spike density time series per trial. These trial-based time series were averaged separately for hits and misses. Finally, a normalization procedure according to *Ison et al., 2015* was carried out to account for the vast variability of firing rates between neurons (some neurons fire very sparsely whereas other fire at a very high rate). Normalized firing density was calculated according to the formula below, where z (t) = normalized firing rate, sd(t) = spike density, μ(bl)= mean spike density in baseline interval, σ(bl) = standard deviation of spike density in baseline interval, and $\lambda$ = regularization parameter (set to 0.1; see *Ison et al., 2015*). This regularization parameter was necessary to avoid extreme values for cases where no or only few spikes were present in the baseline. The baseline interval was set to −1000 ms to −125 ms (i.e. half of the Gaussian window length).

$$z(t) = \frac{sd(t) - \mu(bl)}{\sigma(bl) + \lambda}$$

## LFP pre-processing

The continuous raw data was imported using the Neuralynx data reader provided by U. Rutishauser (7[th] release, https://www.urut.ch/new/serendipity/index.php?/pages/nlxtomatlab.html). A Butterworth low-pass filter was applied (filter order 2) at 300 Hz. In some channels/sessions, an artifact from the TTL pulse was visible, which was removed by subtracting the average artifact from the single trial using a linear regression. Spikes were removed by linearly interpolating the signal from 2 ms before the spike to 6 ms after the spike (*Jacobs et al., 2007*). Line noise at 50 Hz was removed using a template subtraction method. This method estimates the line noise signal by fitting a sinusoid at noise frequency and then subtracting it from the signal. Because the line noise can be assumed to be stationary (whereas brain signals are not stationary), this approach effectively removes line noise while retaining physiological activity. This approach is therefore preferable to a band-stop filter, which does not retain physiological activity at line noise frequency. Finally, the continuous LFP data was segmented into epochs of 14 s duration centered at cue onset during encoding (i.e. animal image), downsampled to 1 khz using Fieldtrip (*Oostenveld et al., 2011*), and stored for further analysis.

The LFP data was cleaned from artifacts and re-referenced as follows. For artifact rejection, a two-step procedure was carried out which first identified noisy channels and then noisy trials. To reject noisy channels, the root mean square amplitude (RMSA) in the time window of interest (−0.5 to 5 s around cue onset) was calculated for each channel and z-transformed (where the mean and standard deviation were obtained across channels). Channels with a z-value above 3 were rejected. On the trial level, a similar procedure was carried out on those channels retained after the first step. The RMSA was averaged in the time window of interest (−0.5 to 5 s) and z-transformed across trials. Trials with a z-value of over 4 were rejected. Furthermore, z-scores were calculated for the raw amplitude. Trials with a maximum raw amplitude z-score of above 4 were also rejected. Lastly, only channels with a minimum of 25 remaining trials were submitted to further analysis.

Finally, the data were re-referenced to yield a comparable signal across patients and to extract the field potential on a very local level. To this end, each microwire channel was re-referenced to the mean amplitude of its neighboring microwire channels (i.e. wires on the same bundle). Bundle refers to the eight high-impedance contacts within one Behnke-Fried hybrid electrode. Instead of simply subtracting the average signal, however, we used an orthonormalization procedure termed Gram-Schmidt Correction (see https://en.wikipedia.org/wiki/Gram–Schmidt_process). This procedure projects two signals into a sub-space where both signals don't share any variance, i.e., they are orthonormal to each other (their sum = 0). This is an effective way of subtracting those parts of a signal, which are shared between an individual channel and its neighbors (such as volume conduction; see *Hipp et al., 2012*). Furthermore, by only subtracting those parts of the signal that are shared between

the two sources this method is less prone to introducing noise from the reference into the referenced channel.

## Spike-field coupling

The following procedure was used to calculate SFC and assess differences in SFC between hits and misses. As an overview, this procedure calculated SFCs for every possible pairing of units (spikes) and LFP channel. Therefore, a given pairing could either constitute coupling at the local level (i.e. spike and LFP come from the same channel, or from the same bundle), or coupling at a distal level (i.e. spike and LFP come from different bundles; see *Figures 2 and 3*). Each pair was first tested for significant SFC and then submitted to further analysis to compare SFC between hits and misses.

As a first step, phase was calculated using Fieldtrip. Different parameters were used for low- (2–40 Hz) and high-frequency ranges (40–80 Hz); i.e., using a 6-cycle wavelet for low frequencies in steps of 1 Hz and a 12-cycle wavelet for high frequencies in steps of 2 Hz. We used these different parameter settings for the two frequency to best capture their temporal dynamics and to deal with data volume. The exact cut-off frequency of 40 Hz was chosen because of a previous paper from our group where we showed that the fast gamma oscillation (between 50 and 80 Hz) in the hippocampus is specifically related to memory encoding (*Griffiths et al., 2019*). In addition, for high frequencies only the first derivative was taken before calculating phase. Phases at spike times were extracted and split according to whether they occurred at the time window of interest, which was 1 s after the face/place stimuli were shown (2–3 s after Cue onset). If at least 30 spikes were available, then the data was admitted to the next step. For a given spike-LFP pair, a Raleigh test using the Circ_Stat Toolbox (http://www.jstatsoft.org/v31/i10) was calculated at each frequency to assess whether phase distributions were different from a uniform distribution. An FDR correction (*Benjamini and Hochberg, 1995*) was applied to correct for multiple comparisons across frequencies. If this corrected p-value was below 0.05, then the spike-LFP pair was submitted to the next step. In this step, PPC was calculated for three conditions, (i) across all trials (hits and misses together), (ii) only for hits (i.e. later completely remembered trials), and (iii) only for misses (i.e. partially remembered and fully forgotten trials).

To statistically assess whether the number of significant spike–LFP couplings identified in the above procedure was above chance a randomization test was carried out. This test was run separately for locally coupled and distally coupled spike–LFP pairs. To this end, trials for the spike-providing data and phase- (i.e. LFP) providing data were shuffled. All other parameters (including number of hit and miss trials) were held constant. These shuffled data were then subjected to the same Raleigh test (FDR-corrected) as above. For each possible spike-LFP pair, 100 such shuffled tests were carried out, and the data was stored. From these shuffled data, a distribution of the number of 'significant' spike-LFP pairs (i.e. Raleigh test p<0.05; FDR-corrected) was generated by drawing 10,000 samples, counting the number of significant spike-LFP pairs for each sample.

To extract the peak frequency of SFC, a peak detection on the individual PPC spectra was run using the findpeaks command in Matlab. For theta, the PPC spectrum was restricted to a frequency range between 3 and 13 Hz and for gamma between 45 and 75 Hz. The peak PPC had to surpass a threshold of 0.005 for both hits and misses to ensure that meaningful PPC was present in both conditions.

## Non-stationarities of LFP

One may be concerned about non-stationarities of the signal and whether such non-stationarities affect the phase estimations obtained with the here applied wavelet analysis. A specific concern is that the phase obtained from the wavelet analysis is not correct when the instantaneous frequency of the LFP signal does not match the frequency of the wavelet. This concern was addressed in two ways.

In a first step, a simulation was carried out where an oscillator in the LFP signal transitions randomly between frequencies with a mean frequency of 6 Hz fluctuating between 3.5 and 9 Hz, thus exhibiting strong non-stationaries across a wide range. Spikes of a hypothetical neuron were simulated to be locked to the trough of this non-stationary oscillator (see *Figure 2—figure supplement 2A-B*). White noise was added to the simulated signal. The results of this simulation demonstrate that the mean frequency of spike phase locking at 6 Hz is correctly retrieved by the wavelet analysis (Figure S6C). Crucially, the mean phase of spiking is also correctly retrieved (Figure S6D). The same simulation has been carried out for the high-frequency range with similar results. From this we can conclude that our

wavelet-based approach is well suited to recover the mean frequency and mean phase of coupling between a spiking neuron and a non-stationary oscillator.

In a second step, we repeated the spike-LFP coupling analysis using a combination of band-pass filter and Hilbert transformation. The band-pass filter (Butterworth) was set to a width of 4 Hz for the low-frequency range and 8 Hz for the high-frequency range, thus leaving ample opportunity for variations in instantaneous frequency. Trials were segmented as before to 14 s centered at the onset of the cue stimulus (i.e. animal) to ensure that filter artifacts at the beginning and end of the trial are well out of the time window of interest (i.e. 2–3 s after cue onset). The band-pass filter was implemented using the ft_preprocessing command in fieldtrip centered at discrete steps of 1 Hz between 3 and 40 Hz (i.e. 3 = 1–5 Hz; 4 = 2–6 Hz; … 40 = 38–42 Hz) and 2 Hz between 40 and 80 Hz (i.e. 40 = 36–44 Hz; 42 = 38–46 Hz; … 80 = 76–84 Hz). Phase was estimated using the Hilbert transform as implemented in fieldtrip (ft_preprocessing), and phase variance was obtained using PPC as described above. The results of this analysis are reported in *Figure 2—figure supplement 3*, *Figure 3—figure supplement 2*.

## Directionality of distal theta SFC

To assess the direction of information between spikes and distally coupled LFPs, the PSI was applied (*Nolte et al., 2008*). The PSI is a frequency-resolved measure to discern the direction of information flow between two neuronal regions or time series (A and B). Like many other directional coupling measures, the PSI assumes a time delay for a signal to travel from A to B. If the speed of travel is constant, then the phase difference between sender and recipient increases with frequency, and a positive slope of the phase spectrum can be expected. Hence, a positive phase slope indicates that A is the sender and B is the receiver, and vice versa. We preferred the PSI over Granger causality as it requires less assumptions and is less sensitive to noise (*Nolte et al., 2008*). The PSI was calculated using the ft_connectivityanalysis function in fieldtrip using a bandwidth of 5 Hz (i.e. phase slopes are estimated over a 5-Hz window). Spike time series were convolved with a Gaussian window (25 ms) to yield a continuous spike density signal. The raw PSI values were baseline corrected using a z-transformation to ensure an unbiased comparison between hits and misses. To this end, trials for the spike-providing signal were shuffled 100 times and submitted to the same PSI analysis as the real data. This shuffling was carried out separately for hits and misses, thereby keeping the number of overall spikes constant. The mean and standard deviation across shuffles were calculated and used for z-transformation of the PSI values of the real (i.e. un-shuffled) data. Finally, a minimum of 60 spikes was imposed to allow for a meaningful analysis of directional information flow between the two signals (see *Figure 4—figure supplement 2* for a control analysis with a minimum of 30 spikes which obtained similar results to those shown in *Figure 4*).

## SFC control analysis for selection bias

In the above analysis, only spike-LFP pairs, which show significant phase coupling across all trials are subjected to further analysis. This could introduce a potential bias, especially for cases where firing rates and trials are not evenly distributed between conditions (hits vs. misses). This concern would be especially problematic if the results would show an overall increase in SFC for hits vs. misses, as opposed to differences in peak frequencies as shown here. Nevertheless, we accounted for this issue by means of a control analysis. In this control analysis, PPC values for a given spike-LFP pair were calculated for 5000 instances where the trials for spike-providing and LFP-providing data were shuffled. As above, all other parameters, most importantly number of hit and miss trials, were kept constant. After each shuffle, a Raleigh test as above was carried out across all trials. If this test retained a p-value <0.05 (FDR-corrected), PPC spectra were calculated for 'pseudo-hits' and 'pseudo-misses'. These spectra therefore contain the same selection bias as in the real analysis because the ratio between hits and misses has been kept constant, and therefore provide a baseline for hits and misses which can be used to effectively correct for this bias. Accordingly, PPC values for each spike-LFP pair for the real data were z-transformed using this baseline by subtracting the mean across 'significant' shuffle runs and dividing by the standard deviation. This bias corrected data was submitted to the same peak detection analysis as the real data, and similar results were obtained (see *Figure 2—figure supplement 1*, *Figure 3—figure supplement 1*).

## Theta-gamma interaction

To assess the spatial overlap of local SFC in the gamma range and distal SFC in the low frequency range, the number of spike-LFP pairs was counted that exhibited both phenomena in the same region (i.e. same microwire bundle). CFC was then analyzed for these overlapping pairs of locally and distally coupled spike-LFP channels. To this end, both theta and gamma LFPs were taken from the same bundle of microwires and therefore were recorded in the same region (or at least in very close proximity; see *Figure 4A-B*). The phase amplitude coupling (PAC) analysis therefore does reflect the temporal coordination of theta and gamma oscillations in a local region. It is only that the locally recorded theta is phase locked to a spike recorded in another, distally coupled, region. For each channel pair, the peak gamma frequency and the peak theta frequency were extracted from the PPC spectra. Importantly, these peaks were extracted separately for hits and misses to account for the difference in frequency between the two conditions. Peaks at gamma were restricted to a frequency range between 50 and 80 Hz, and 5 and 11 Hz for theta. Phase for the lower frequencies (i.e. theta) and power for the higher frequencies (i.e. gamma) were calculated using the same wavelet parameters as above (i.e. 6 cycles for theta, 12 cycles for gamma). CFC was then calculated via the MI (*Tort et al., 2010*) using the function 'ModIndex_v2.m' provided by Adriano Tort with a binning parameter of 18 bins (https://github.com/tortlab/phase-amplitude-coupling; *Tort, 2018*). The MI was calculated separately for hits and misses. Because the MI is affected by trial numbers, a normalization procedure was applied to yield a bias-free CFC measure for hits and misses. To this end, trials for the phase-providing channel were shuffled, and MI was calculated after each shuffle (N=200). This was done separately for each condition (hits and misses) to generate a baseline MI under the null for each condition. The mean and standard deviation across the shuffled data were then used to z-transform the MI of the real data.

## Power analysis

Power of LFP data was calculated using the same wavelet parameters for high- and low-frequency ranges as above. Raw power values were z-transformed for each channel separately using a common procedure for analyzing subsequent memory effects (*Griffiths et al., 2019*; *Sederberg et al., 2007*; *Burke et al., 2013*). Power values for each channel, frequency band, and trial, were first averaged across time (−0.5 to 5 s). Then the median and standard deviation of this time-averaged power across trials were calculated. Power values for each channel, frequency bin, and trial were then z-transformed by subtracting median power and dividing by standard deviation. Trials containing outlier power values (i.e. maxima >2.5 of standard deviation of maxima across trials) were discarded. Finally, trials were split according to conditions (hits and misses) and averaged. Power for low frequencies was calculated for channels which showed a significant SFC in the low-frequency range, whereas power for high frequencies was calculated only for those channels showing significant SFC in the high frequencies.

An additional power analysis was carried out to assess the presence of a meaningful signal at the frequencies where spike-LFP coupling was observed (see *Figure 2—figure supplement 4*, *Figure 3—figure supplement 3*). To this end, power at spike times was extracted for distally coupled LFPs in the low-frequency range and locally coupled LFPs in the gamma frequency range using the same wavelet filter as used for SFC analysis. Power spectra were 1/f corrected by fitting and subtracting a linear function to the log-log transformed power spectra. The resulting spectra were then back transformed to linear space and centered at the peak frequency of PPC. Lastly, power values were normalized for each LFP channel by subtracting the mean across the whole spectrum and dividing by the standard deviation across the whole spectrum.

## Inter-trial phase consistency

Inter-trial phase consistency was calculated for the lower frequency range (i.e. 2–40 Hz) using PPC (*Vinck et al., 2010*). Phase for each single trial was extracted using the same wavelet transformation (six cycles) as above. PPC spectra were limited to a frequency range of interest (2–13 Hz), comprising the slow and fast theta range. PPC spectra were calculated for hits and misses separately and only for those channels showing significant SFC in the low-frequency ranges.

## Spike power analysis

To test for the presence of theta rhythmicity within the spiking of neurons, an FFT analysis was carried out on the spike density time series. Spike time series were convolved with a Gaussian window (25 ms) to yield a continuous spike density signal for the time window of interest (2–3 s after cue onset). These trials where then submitted to an FFT analysis using the ft_freqanalysis command in fieldtrip, with a Hanning taper and frequency smoothing of ± 2 Hz. Results are shown in Figure S9.

## Theta waveshape asymmetry

To assess the presence of asymmetric theta waveforms, which could give rise to spurious CFC at harmonics of theta, two control analyses were performed. For the first control analysis, MIs were calculated exactly the same way as for the real data, except that for each LFP signal the gamma frequency was taken as the eighth harmonic of the theta in that signal (i.e. for theta = 9 Hz, gamma = 9 × 8 = 72 Hz). We took the eighth harmonic as this was the frequency that was on average closest to the observed gamma frequency (i.e. 64–72 Hz). The eighth harmonic was estimated based on the dominant theta frequency separately for hits and misses. Next, we calculated the difference in MI between hits and misses. If asymmetric theta gives rise to the difference in CFC the real data then this effect should even be stronger in the control data where gamma was centered at the harmonic of theta. However, we observed the opposite pattern, with the real data showing stronger differences between hits and misses compared to the harmonic control data (*Figure 5—figure supplement 1A-B*).

In a second control analysis, the asymmetry index (AI) was calculated following the procedure described in *Belluscio et al., 2012*. To this end, the LFP data for hits and misses was filtered at the theta phase providing frequency band ± 2 Hz used for CFC analysis, using a Butterworth bandpass filter implemented in Fieldtrip. Then the filtered data was cut to the time window of interest (2–3 s), and peaks and troughs were identified using the 'findpeaks' function in Matlab. Thereafter, peaks and troughs of the filtered data were adjusted by matching them with peaks and troughs in the unfiltered data. This matching process searched for local maxima and minima in a time window of a quarter cycle length of the theta frequency (i.e. 5 Hz = 200/4 = 50 ms) centered at the peak located in the filtered data. The time stamps of these adjusted peaks and troughs were then used to calculate the asymmetry of the theta waveshape using the formula below. Where *Tasc* is the duration of the ascending flank, *Tdesc* is the duration of the descending flank, and $\omega$ is the cycle length of the theta frequency. *Tasc* was calculated by subtracting the time stamp of the trough from its subsequent peak (i.e. T peak(t+1) – T trough(t)); Tdesc was calculated by subtracting the time stamp of the peak from its subsequent trough (i.e. T trough(t+1) – T peak(t)). Therefore, both Tasc and Tdesc always yield positive values. The AI is a normalized measure ranging between −1 and 1 with negative values indicating longer durations of descending flanks, and positive values indicating longer durations for ascending flanks.

$$AI = \frac{Tasc - Tdesc}{\omega}$$

To verify that this index detects the presence of asymmetric waveforms, a dataset recorded in the rodent entorhinal cortex was used as a positive control (courtesy of Ehren Newman). The data comprised 10 min of open field navigation of the rodent. Clear ongoing theta activity was present in this data and indeed a strong asymmetry could be detected (Figure S10C). For the human data, AIs were averaged across trials for hits and misses separately to yield one AI per channel. AIs between hits and misses were compared using a paired samples T-test. No difference in asymmetry between hits and misses was obtained (see Figure S10D), which further rules out an influence of asymmetric waveshapes on the observed CFC results.

## Co-firing analysis

Co-firings between pairs of single/multi-units (referred to as units in the below) at different time lags were calculated via cross-correlations using the xcorr function in Matlab. To this end, spike time series for each unit were concatenated across trials to yield one vector and convolved with a Gaussian envelope with a width of 25 ms (~10 ms full width half maximum). We chose this time window because it should represent a good balance between integrating over a long-enough time window and thus allowing for some jitter in neural firing between pairs of neurons, while still being temporally specific (*Cohen and Kohn, 2011*). To test whether this choice critically affected our results, we repeated the

analysis for different window sizes, i.e., 15, 35, and 45 ms (see *Figure 6—figure supplement 4*). Cross-correlations were calculated for pairs of putative 'sending' units, and putative 'receiving' units. Putative sending units were units, which showed a significant distal SFC in the low-frequency range, and where the distally coupled LFP was in addition locally coupled to a unit in the high-frequency (gamma) range. This locally gamma-coupled unit was taken as the putative receiving unit. Accordingly, the pairs were chosen such that the region where the sending unit was coupled to is the same as the region where the local coupling occurs. This resulted in a relatively low number of neural pairs (N=24), because two conditions needed to be met. More specifically, for a pair of units to be considered as such the putative sending unit needed to show significant distal low-frequency coupling to the region where the putative receiving unit was located; additionally, the receiving unit needed to be significantly coupled to local gamma oscillations. This selection and labeling of units into sending and receiving units are supported by the directional coupling analysis (see *Figure 4*). Altogether, five patients contributed data to this analysis, with a median of five pairs per patient, and a minimum and maximum of two and eight pairs per patient, respectively (see *Figure 6—source data 1*).

Cross-correlations between putative sending and putative receiving units were calculated separately for hits and misses. To correct for a potential bias of numbers of spikes, cross-correlation values for hits and misses were z-transformed according to a shuffled baseline. To this end, trials for the putative receiving unit were shuffled 2000 times and submitted to the same cross-correlation analysis as the real data. This shuffling was carried out separately for hits and misses, thereby keeping the number of overall spikes constant. The mean and standard deviation across shuffles were calculated and used for z-transformation of the cross-correlation values of the real (i.e. un-shuffled data). As a final step, only cross-correlations for pairs of neurons where the average co-incidence (mean across hits and misses) exceeded 1 at any lag were admitted to statistical analysis. This step ensured that only pairs of neurons with meaningful co-firings were used. To compare the latencies of co-firing between hits and misses a peak detection was carried out using the 'findpeaks' function in Matlab.

## Acknowledgements

We like to thank all patients for taking time to participate in the experimental sessions. We also thank Markus Siegel for providing useful comments on previous versions of the manuscript and Pieter Roelfsema for supporting us with data collection. SH was supported by grants from the European Research Council (Nr. 647954), and the Economic and Social Research Council (ES/R010072/1). MtW and MW were supported by a grant from the European Research Council (StG-715714).

## Additional information

### Funding

| Funder | Grant reference number | Author |
|---|---|---|
| European Research Council | 647954 | Simon Hanslmayr |
| Economic and Social Research Council | ES/R010072/1 | Simon Hanslmayr |

The funders had no role in study design, data collection and interpretation, or the decision to submit the work for publication.

### Author contributions

Frédéric Roux, Conceptualization, Data curation, Software, Formal analysis, Investigation, Methodology, Writing - original draft, Project administration, Writing - review and editing; George Parish, Resources, Software, Methodology, Writing - review and editing; Ramesh Chelvarajah, Hajo Hamer, Bernhard P Staresina, Resources, Methodology, Project administration, Writing - review and editing; David T Rollings, Resources, Data curation, Methodology, Project administration; Vijay Sawlani, Stephanie Gollwitzer, Gernot Kreiselmeyer, Resources, Methodology, Project administration; Marije J ter Wal, Resources, Data curation, Software, Methodology, Writing - review and editing; Luca Kolibius,

Resources, Investigation, Writing - review and editing; Maria Wimber, Data curation, Project administration, Writing - review and editing; Matthew W Self, Software, Methodology, Writing - review and editing; Simon Hanslmayr, Conceptualization, Data curation, Software, Formal analysis, Supervision, Funding acquisition, Validation, Investigation, Visualization, Methodology, Writing - original draft, Project administration, Writing - review and editing

## Author ORCIDs
George Parish http://orcid.org/0000-0002-7533-8298
Marije J ter Wal http://orcid.org/0000-0003-4922-3435
Bernhard P Staresina http://orcid.org/0000-0002-0558-9745
Maria Wimber http://orcid.org/0000-0002-1917-353X
Matthew W Self http://orcid.org/0000-0001-5731-579X
Simon Hanslmayr http://orcid.org/0000-0003-4448-2147

## Ethics
Informed consent to participate in the experiments and consent to publish the results was obtained from the patients prior to data collection. Ethical approvals were given by National Research Ethics Service (NRES), Research Ethics Committee (Nr. 15/WM/0219), the ethical review board of the Friedrich-Alexander Universität Erlangen-Nürnberg (Nr. 124_12 B), and the Medical Ethical Review board of the Vrije Universiteit Medisch Centrum (Nr. NL55554.029.15), for Birmingham, Erlangen and Amsterdam respectively.

## Decision letter and Author response
Decision letter https://doi.org/10.7554/eLife.78109.sa1
Author response https://doi.org/10.7554/eLife.78109.sa2

## Additional files

### Supplementary files
• Transparent reporting form

### Data availability
All code used for data analysis and visualization of results is deposited here: https://osf.io/fngz8/.

The following dataset was generated:

| Author(s) | Year | Dataset title | Dataset URL | Database and Identifier |
|---|---|---|---|---|
| Hanslmayr S, Roux F | 2022 | Oscillations support short latency co-firing of neurons during human episodic memory formation | https://doi.org/10.17605/OSF.IO/FNGZ8 | Open Science Framework, 10.17605/OSF.IO/FNGZ8 |

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
