## [Editor Report]

Roux and colleagues measured spiking activity and local field potentials predominantly from the hippocampus and also a few surrounding structures in the medial temporal lobe from patients with pharmacologically intractable epilepsy while the patients performed an associative memory task. Their data are convincing and provide important insights into how neurons in the medial temporal lobe correlate with associative memory.

---

## [Decision Letter]

**Decision letter after peer review:**

Thank you for submitting your article "Oscillations support short latency co-firing of neurons during human episodic memory formation" for consideration by *eLife*. Your article has been reviewed by 2 peer reviewers, and the evaluation has been overseen by a Reviewing Editor and Laura Colgin as the Senior Editor. The following individuals involved in the review of your submission have agreed to reveal their identity: Gabriel Kreiman (Reviewer #1); Taufik Valiante (Reviewer #2).

Essential revisions:

Most points raised by the reviewers concerned methodological choices, clarifications (e.g.) regarding what figures 2-6 convey and which data i.e., electrodes/area are presented, and justification of some of the parameters chosen (e.g., how frequency bands are defined). Specific requests can be found within the reviews of both reviewers 1 and 2.

*Reviewer #1 (Recommendations for the authors):*

First, I must apologize to the editors and authors for the delay in reviewing this manuscript. This should not be interpreted as a lack of interest in the work. On the contrary, I was excited to read this manuscript but life and many other commitments got in the way. It is very frustrating always for editors to chase reviewers and for authors to await for reviews for prolonged periods of time.

Figures 2 through 6. There is no description of the relationship between the findings and the anatomical location of the electrodes (other than distal versus local). Perhaps the non-uniform distribution of electrodes makes these analyses more complicated and such questions might have minimal if any statistical power. But how should we think about the claims in Figures 2-6 in relationship to the hippocampus, amygdala, entorhinal cortex, and parahippocampal gyrus? As one example question out of many, is Figure 2C revealing results for local pairs in all medial temporal lobe areas or any one area in particular? I won't spell out every single anatomical question. But essentially every figure is associated with an anatomical question that is not described in the results.

Figure 1

1A. I assume that image positions are randomized during a cued recall?

What was the correlation between subjects' indication of how many images they thought they remembered and their actual performance?

1B. Chance is shown for hits but not misses. I assume that hits are defined as both images correct and misses as either 0 or 1 image correct. Then a chance for misses is 1-chance for hits = 5/6. It would be nice to mark this in the figure.

The authors report that both incorrect was 11.9%. By chance, both incorrect should be the same as both correct, hence also 1/6 probability, hence the probability of both incorrect seems quite close to chance levels, right?

1C. How does the number of electrodes relate to the number of units recorded in each area?

Line 152. The authors state that neural firing during encoding was not modulated by memory for the time window of interest. This is slightly surprising given that other studies have shown a correlation between firing rates and memory performance (see Zheng et al. Nature Neuroscience 2022 for a recent example). The task here is different from those in other studies, but is there any speculation as to potential differences? What makes firing rates during encoding correlate with subsequent memory in one task and not in another? And why is the interval from 2-3 seconds more interesting than the intervals after 3 seconds where the authors do report changes in firing rates associated with subsequent performance? Is there any reason to think that the interval from 2-3 seconds is where memories are encoded as opposed to the interval after 3 seconds?

Lines 154-157 and relationship to the subsequent analyses. These lines mention in passing differences in power in low-frequency bands and high-frequency bands. To what extent are subsequent results (especially Figures 3 and 4) related to this observation? That is, are the changes in spike-field coherence, correlated with, or perhaps even dictated by, the changes in power in the corresponding frequency bands?

Do local interactions include spike-field coherence measurements from the same microwire (i.e., spikes and LFPs from the same microwire)?

60 Hz. It has always troubled me deeply when results peak at 60 Hz. This is seen in multiple places in the manuscript; e.g., Figures 2B, 2E. What are the odds that engineers choosing the frequency for AC currents would choose the exact same frequency that evolution dictated for interactions of brain signals? This is certainly not the only study that reports interesting observations peaking at 60 Hz. One strong line of argument to suggest that this is not line noise is the difference between conditions. For example, in Figure 2B, there is a difference between local and distal interactions. It is hard for me to imagine why line noise would reveal any such difference. Still …

Figure 6. I was very excited about Figure 6, which is one of the most novel aspects of this study. In addition to the anatomical questions about this figure noted above, I would like to know more. What is the width of the Gaussian envelope? Are these units on the same or different microwires? How do the spike latencies reported here depend on the firing rates of the two units? What do these results look like for other pairs that are not putative upstream/downstream pairs?

*Reviewer #2 (Recommendations for the authors):*

1. In a number of places a causal role for oscillations is suggested, ie: "to analyze how brain oscillations promote co-incidences of firing between neurons during successful and unsuccessful encoding of episodic memories" and the last sentence of the abstract also. That oscillations "promote" feels a bit strong, especially in light of recent work like (Schneider et al. 2021. "A Mechanism for Inter-Areal Coherence through Communication Based on Connectivity and Oscillatory Power." Neuron 109 (24): 4050-67.e12.) and (Rolls, E. T., T. J. Webb, and G. Deco. 2012. "Communication before Coherence." The European Journal of Neuroscience 36 (5): 2689-2709.). Might be good to relax this a bit?

2. Line 80: "Therefore, considering the fast decay of the membrane potential, a neural assembly which synchronizes firing at faster oscillations is more likely to drive a downstream neural assembly compared to synchronization at slower oscillations. " Is not made with any justification, and seems to be an important concept for this paper. Would be good to unpack this for the reader and provide justification for this.

3. Line 425: This is a really nice interpretation of the findings of how excitability changes the frequency of the theta and γ oscillations and seems plausible. This reference (Lefebvre et al. 2015. "Stimulus Statistics Shape Oscillations in Nonlinear Recurrent Neural Networks." The Journal of Neuroscience: 35 (7): 2895-2903.) might also be a nice to that supports this.

4. Line 695 & 725: Why were the frequency ranges of low (2-40), and high (40-80) defined as such. As well PPC spectrum is restricted to 3-13Hz, and 45-75Hz for γ. Although admittedly the precise ranges are always open to debate theta is typically considered 4-8Hz and γ 30-60, with high γ above 60. The choice of frequency range will likely change the statistics for example in Figure S4 then the stats are false discovery rate corrected. Can the authors justify the frequency ranges used for these analyses?

5. Line 760: The text to describe PSI seems very similar to some of the text in Nolte, and would suggest revising a bit.

6. 770: A minimum of 60 spikes is used here, and for SFC 30. Why the difference, and how are these justified.

7. 798: CFC can also be artificially elevated as would be SFC by broad band power introduced by incomplete spike removal and/ or multiunit activity. Can the authors show that such an artifact does not underlie the CFC measures?

[Editors' note: further revisions were suggested prior to acceptance, as described below.]

Thank you for resubmitting your work entitled "Oscillations support short latency co-firing of neurons during human episodic memory formation" for further consideration by *eLife*. Your revised article has been evaluated by Laura Colgin (Senior Editor) and a Reviewing Editor.

The manuscript has been significantly improved and most of the queries posted by the reviewers have been addressed satisfactorily. There are some remaining issues that might still need to be addressed, as outlined below:

– Both reviewers were concerned with the contaminations of spikes when analysing data from the same electrode. Those were dealt with very well with analysis considering 'silent' electrodes and also by removing those electrodes contributing spikes and LFPs. As such the issue has been dealt with, yet, it might be best to present the analysis in the full paper without those confounds and present in the supplement the full dataset.

– Schneider et al. 2021. Neuron has provided an important alternative explanation to the findings of oscillations as a mechanism for inter areal communication. While the authors have toned down a causal interpretation of oscillations, it would be best to discuss in further length the paper by Schneider.

---

## [Author Response]

Essential revisions:Most points raised by the reviewers concerned methodological choices, clarifications (e.g.) regarding what figures 2-6 convey and which data i.e., electrodes/area are presented, and justification of some of the parameters chosen (e.g., how frequency bands are defined). Specific requests can be found within the reviews of both reviewers 1 and 2.Reviewer #1 (Recommendations for the authors):First, I must apologize to the editors and authors for the delay in reviewing this manuscript. This should not be interpreted as a lack of interest in the work. On the contrary, I was excited to read this manuscript but life and many other commitments got in the way. It is very frustrating always for editors to chase reviewers and for authors to await for reviews for prolonged periods of time.Figures 2 through 6. There is no description of the relationship between the findings and the anatomical location of the electrodes (other than distal versus local). Perhaps the non-uniform distribution of electrodes makes these analyses more complicated and such questions might have minimal if any statistical power. But how should we think about the claims in Figures 2-6 in relationship to the hippocampus, amygdala, entorhinal cortex, and parahippocampal gyrus? As one example question out of many, is Figure 2C revealing results for local pairs in all medial temporal lobe areas or any one area in particular? I won't spell out every single anatomical question. But essentially every figure is associated with an anatomical question that is not described in the results.

To address the reviewer’s point we now report the distribution of spike-LFP pairs across anatomical regions for each Figure 2-6. The results split by anatomical regions are reported in Figure 2 —figure supplement 7, Figure 3 —figure supplement 7, Figure 4 —figure supplement 1, Figure 5 —figure supplement 2, and Figure 6 —figure supplement 3. We also calculated a non-parametric Kruskal-Wallis Test to statistically examine the effect of anatomical regions on the results shown in each figure. Generally, these new results show that the effects are similar across regions, apart from two exceptions (i.e. Figure 4 – supplement 1; and Figure 5 – supplement 2). However, we would like to stress that these results should be taken with a huge grain of salt because the electrodes were not evenly distributed across regions (i.e. ~75% of observations pertain to the hippocampus), and patients as the reviewer correctly points out. This leads to sometimes very low numbers of observations per region and it is difficult to disentangle whether any apparent differences are driven by regional differences, or differences between patients. Detailed results are reported below.

Manuscript lines 207-212: “In the above analysis all MTL regions were pooled together to allow for sufficient statistical power. Results separated by anatomical region are reported in Figure 2 —figure supplement 7 for the interested reader. However, these results should be interpreted with caution because electrodes were not evenly distributed across regions and patients making it difficult to disentangle whether any apparent differences are driven by actual anatomical differences, or idiosyncratic differences between patients.”

Manuscript lines 255-258: “Finally, we report the distal spike-LFP results separated by anatomical region in Figure 3 —figure supplement 7, which did not reveal any apparent differences in the memory related modulation of theta spike-LFP coupling between regions.”

Manuscript lines 264-266: “PSI results separated by anatomical regions are reported in Figure 4 —figure supplement 1, which revealed that the PSI results were mostly driven by within regional coupling.”

Manuscript lines 399-303: “We also analyzed whether the memory-dependent effects of cross-frequency coupling differ between anatomical regions (see Figure 5 —figure supplement 2). This analysis revealed that the results were mostly driven by the hippocampus, however we urge caution in interpreting this effect due to the large sampling imbalance across regions.”

Manuscript lines 343-346: “As for the above analysis we also investigated any apparent differences in co-firing between anatomical regions. These results are reported in Figure 6 —figure supplement 3 and show that the earlier co-firing for hits compared to misses was approximately equivalent across regions.”

Figure 11A. I assume that image positions are randomized during a cued recall?

Yes, that was the case. We now added that information in the methods section.

Manuscript lines 526: “Image positions on the screen were randomized for each trial.”

What was the correlation between subjects' indication of how many images they thought they remembered and their actual performance?

We did not log how many images the patients thought they remembered. Specifically, if the patients answered that they remembered at least one image, then they were shown the selection screen where they could select the appropriate images. Therefore, we cannot perform this analysis. We report this now in the methods section. However, albeit interesting, the results of such an analysis would not affect the main conclusions of our manuscript.

Manuscript lines 523-524: “The experimental script did not log how many images the patient indicated that they thought to remember.”

1B. Chance is shown for hits but not misses. I assume that hits are defined as both images correct and misses as either 0 or 1 image correct. Then a chance for misses is 1-chance for hits = 5/6. It would be nice to mark this in the figure.

Done as suggested (see new Figure 1).

The authors report that both incorrect was 11.9%. By chance, both incorrect should be the same as both correct, hence also 1/6 probability, hence the probability of both incorrect seems quite close to chance levels, right?

Yes, that is correct, however, across sessions the proportion of full misses (i.e. both incorrect) was significantly below chance (t(39)=-1.9214; p<0.05). Nevertheless, the proportion of fully forgotten trials appears to be higher than expected purely by chance. This is likely driven by a tendency of participants to either fully remember an episode, or completely forget it, as demonstrated previously in behavioural work (Joensen et al., 2020; JEP Gen.). We report this now in the manuscript.

Manuscript lines 132-136: “Across sessions the proportion of full misses (i.e. both incorrect) was significantly below chance (t_39_=-1.92; p<0.05). However, the proportion of fully forgotten trials appears to be higher than expected purely by chance. This is likely driven by a tendency of participants to either fully remember an episode, or completely forget it, as demonstrated previously in behavioral work (25).”

1C. How does the number of electrodes relate to the number of units recorded in each area?

The distribution of neurons per region is shown in the new Figure 1D (see above). It approximately matches the distribution of electrodes per region, except for the Amygdala where slightly more neurons where recorded. This is because of one patient (P08) who had two electrodes in the left and right Amygdala and who contributed at lot of sessions (i.e. 9 sessions, comparing to an average of 4.44 per patient).

Line 152. The authors state that neural firing during encoding was not modulated by memory for the time window of interest. This is slightly surprising given that other studies have shown a correlation between firing rates and memory performance (see Zheng et al. Nature Neuroscience 2022 for a recent example). The task here is different from those in other studies, but is there any speculation as to potential differences? What makes firing rates during encoding correlate with subsequent memory in one task and not in another? And why is the interval from 2-3 seconds more interesting than the intervals after 3 seconds where the authors do report changes in firing rates associated with subsequent performance? Is there any reason to think that the interval from 2-3 seconds is where memories are encoded as opposed to the interval after 3 seconds?

Zheng et al. used a movie-based memory paradigm where they manipulated transitions between scenes to identify event cells and boundary cells. They show that boundary cells, which made up 7.24% of all recorded MTL cells, but not event cells (6.2% of all MTL cells) modulate their firing rate around an event depending on later memory. There are quite a few differences between Zheng et al’s study and our study that need to be considered. Most importantly, we did not perform a complex movie-based memory paradigm as in Zheng et al. and therefore cannot identify boundary cells, which would be expected to show the memory dependent firing rate modulation. This alone could contribute to the fact that no significant differences in firing rates in the first second following stimulus onset were observed. Such an absence of a difference of neural firing depending on later memory is not unprecedented. In their seminal paper, Rutishauser et al. (2010; Nature) report no significant differences in firing rates (0-1 seconds after stimulus onset, which is similar to our 2-3 seconds time window) between later remembered or later forgotten images. This finding is also in line to Jutras & Buffalo (2009; J Neurosci) who also show no significant difference in firing rates of hippocampal neurons during encoding of remembered and forgotten images.

The 2-3 seconds time interval, which corresponds to 0-1 seconds after the onset of the two associate images, is special because it marks the earliest time point where memory formation can start, therefore allowing us to investigate these very early neural processes that set the stage for later memory-forming processes. While speculative, these early processes likely capture the initial sweep of information transfer into the MTL memory system which arguably is reflected in the timing of spikes relative to LFPs. It is conceivable that these initial network dynamics reflect attentional processes, which act as a gate keeper to the hippocampus (Moscovitch, 2008; Can J Exp Psychol) and thereby set the stage for later memory forming processes. This interpretation would be consistent with studies in macaques showing that attention increases spike-LFP coupling, whilst not affecting firing rates (Fries et al., 2004; Science). We modified the Discussion section to address this issue.

Manuscript lines 468-474: “Interestingly, these early modulations of neural synchronization by memory encoding were observed in the absence of modulations of firing rates, which is consistent with previous results in humans (16) and macaques (12), but contrasts with (43). Studies in macaques showed that attention increases spike-LFP coupling whilst not affecting firing rates (44). It is therefore conceivable that these initial network dynamics reflect attentional processes, which act as a gate keeper to the hippocampus and thereby set the stage for later memory forming processes (45).”

Lines 154-157 and relationship to the subsequent analyses. These lines mention in passing differences in power in low-frequency bands and high-frequency bands. To what extent are subsequent results (especially Figures 3 and 4) related to this observation? That is, are the changes in spike-field coherence, correlated with, or perhaps even dictated by, the changes in power in the corresponding frequency bands?

To address this question we repeated the analysis that we performed for SFC for Power in those channels whose LFP was locally coupled to spikes in γ, and distally coupled to spikes in theta. Furthermore, we correlated the difference in peak frequency between hits and misses between Power and SFC. If power would dictate the effects seen in SFC then we would expect similar effects of memory in power as in SFC, that is an increase of peak frequency for hits compared to misses for γ and theta. Furthermore, we would expect to find a correlation between the peak frequency differences in power and SFC. None of these scenarios were confirmed by the data. These results are now reported in Figure 2 —figure supplement 5 for γ, and Figure 3 —figure supplement 5 for theta.

Manuscript lines 195-199: “We also tested whether a similar shift in peak γ frequency as observed for spike-LFP coupling is present in LFP power, and whether memory-related differences in peak γ spike-LFP are correlated with differences in peak γ power (Figure 2 —figure supplement 5). Both analyses showed no effects, suggesting that the effects in spike-LFP coupling were not coupled to, or driven by similar changes in LFP power.”

Manuscript lines 248-253: “As for γ, we also tested whether a similar shift in peak theta frequency is present in LFP power, and whether there is a correlation between the memory-related differences in peak theta spike-LFP and peak theta power (Figure 3 —figure supplement 5). Both analyses showed no effects, suggesting that the effects in spike-LFP coupling were not coupled to, or driven by similar changes in LFP power.”

Do local interactions include spike-field coherence measurements from the same microwire (i.e., spikes and LFPs from the same microwire)?

Yes, they do. Out of the 53 local spike-SFC couplings found for the γ frequency range, 11 (20.75%) were from pairs where the spikes and LFPs were measured on the same microwire. We assume that the reviewer is asking this question because of a concern that spike interpolation may introduce artifacts which may influence the spectrograms and consequently the spike-LFP coupling measures. This was also pointed out by Reviewer #2. To address this concern, we split the data based on whether the spike and LFP providing channels were the same or different. The results show that (i) the spectrogram of SFC is highly similar between the two datasets, with a prominent γ peak present in both and no significant differences between the two; (ii) restricting the analysis to those data where the LFP and spike providing channels are different replicated the main finding of faster γ peak frequencies for hits compared to misses; and (iii) limiting the SFC analysis further to only ‘silent’ channels, i.e. channels where no SUA/MUA activity was present at all also replicated the main finding of faster γ peak frequencies for hits compared to misses.

These analyses suggest that the SFC results were not driven by spike interpolation artefacts.

Manuscript lines 199-203: “To rule out concerns about possible artifacts introduced by spike interpolation we repeated the above analysis for spike-LFP pairs where the spike and LFP providing channels are the same or different, and for ‘silent’ LFP channels (i.e. channels were no SUA/MUA activity was detected see Figure 2 —figure supplement 6).”

60 Hz. It has always troubled me deeply when results peak at 60 Hz. This is seen in multiple places in the manuscript; e.g., Figures 2B, 2E. What are the odds that engineers choosing the frequency for AC currents would choose the exact same frequency that evolution dictated for interactions of brain signals? This is certainly not the only study that reports interesting observations peaking at 60 Hz. One strong line of argument to suggest that this is not line noise is the difference between conditions. For example, in Figure 2B, there is a difference between local and distal interactions. It is hard for me to imagine why line noise would reveal any such difference. Still …

The frequency for AC currents in Europe is 50 Hz, not 60 Hz as in the US. Therefore, our SFC effects are well outside the range of the notch.

Figure 6. I was very excited about Figure 6, which is one of the most novel aspects of this study. In addition to the anatomical questions about this figure noted above, I would like to know more. What is the width of the Gaussian envelope?

The width of the Gaussian Window used in the original results was 25ms. We chose this time window because in our view it represents a good balance between integrating over a long-enough time window and thus allowing for some jitter in neural firing between pairs of neurons, whilst still being temporally specific. Finding the right balance here is not trivial because a too short time window underestimates co-firing, and a too long time window may not provide the temporal specificity necessary to detect co-firing lags (Cohen & Kohn, 2011; Nat Neurosci). To test whether this choice critically affected our results, we repeated the analysis for different window sizes, i.e. 15, 35, and 45 ms. The results show that the pattern of results did not change, with hits showing earlier peaks in co-firing compared to misses. Critically, the difference in co-firing peaks was significant for all window sizes, except for the shortest one which presumably is due to the increase in noise because of the smaller time window over which spikes are integrated. These issues are now mentioned in the methods section, and the results for the different window sizes are reported in Figure 6 —figure supplement 4.

Manuscript lines 346-347: “The co-firing analyses were replicated with different smoothing parameters (see Figure 6 —figure supplement 4).”

Manuscript lines 894-898: “We chose this time window because it should represent a good balance between integrating over a long-enough time window and thus allowing for some jitter in neural firing between pairs of neurons, whilst still being temporally specific (57). To test whether this choice critically affected our results, we repeated the analysis for different window sizes, i.e. 15, 35, and 45 ms (see Figure 6 —figure supplement 4).”

Are these units on the same or different microwires?

All units used for the analysis shown in Figure 6 come from different microwires. This was naturally the case because the putative up-stream neuron was distally coupled to the theta LFP, and the putative down-stream neuron was locally coupled to γ at this same theta LFP electrode. This information is listed in Figure 6 – source data 1 which lists the locations and electrode IDs for all neuron pairs shown in figure 6.

How do the spike latencies reported here depend on the firing rates of the two units?

To address this question we first tested whether firing rates (averaged across the putative up-stream and down-stream neurons) differ between hits and misses. If they do, this would be suggestive of a dependency of the spike latency differences between hits and misses on firing rates. No such difference was observed (p>0.3). Second, we correlated the differences between hits and misses in Co-firing peak latencies with the differences in firing rates. Again, no significant correlation was observed (R=-0.06; p>0.7), suggesting that firing rates had no influence on the observed differences in co-firing latencies. These control analyses are now reported in the main text.

Manuscript lines 347-350: “No significant differences in firing rates between hits and misses were found (p>0.3), and on correlations between firing rates and the co-firing latencies were obtained (R=-0.06; p>0.7), suggesting that firing rates had no influence on the observed co-firing differences between hits and misses.”

What do these results look like for other pairs that are not putative upstream/downstream pairs?

As we reported in the original manuscript in lines 352-355 we did not find a memory dependent effect on co-firing latencies if we select neuron pairs solely on the basis of distal theta SFC. Within this analysis the distally theta coupled neuron would be the up-stream neuron and the neuron recorded at the site where the theta LFP is coupled would be the down-stream neuron. This null-result suggests that in order for the memory dependent difference in co-firing lags to emerge, the down-stream neurons need to be coupled to a local γ rhythm in order for the memory effect on co-firing latencies to emerge. However, within this previous analysis there is still a notion of up-stream and down-stream neurons because neuron pairs were selected based on distal theta phase coupling. We therefore repeated this analysis for all pairs of neurons in a completely unconstrained fashion such that all possible pairs of neurons that were recorded from different electrodes were entered into the co-firing analysis. This analysis also revealed no difference in co-firing lags, neither for positive lags nor for negative lags. Instead, what this analysis showed is tendency for hits to show a higher occurrence of simultaneous or near simultaneous firing, which is in line with Hebbian learning. These results are now reported in Figure 6 —figure supplement 1.

Manuscript lines 333-335: “In addition, a completely unconstrained co-firing analysis where all pairs possible pairings of units were considered also showed no systematic difference in co-firing lags between hits and misses (Figure 6 —figure supplement 1).”

Reviewer #2 (Recommendations for the authors):1. In a number of places a causal role for oscillations is suggested, ie: "to analyze how brain oscillations promote co-incidences of firing between neurons during successful and unsuccessful encoding of episodic memories" and the last sentence of the abstract also. That oscillations "promote" feels a bit strong, especially in light of recent work like (Schneider et al. 2021. "A Mechanism for Inter-Areal Coherence through Communication Based on Connectivity and Oscillatory Power." Neuron 109 (24): 4050-67.e12.) and (Rolls, E. T., T. J. Webb, and G. Deco. 2012. "Communication before Coherence." The European Journal of Neuroscience 36 (5): 2689-2709.). Might be good to relax this a bit?

We agree with the reviewer and have toned down these statements, especially because we do not provide proof for a causal role of oscillations (see lines 26 and 31-32 in abstract).

2. Line 80: "Therefore, considering the fast decay of the membrane potential, a neural assembly which synchronizes firing at faster oscillations is more likely to drive a downstream neural assembly compared to synchronization at slower oscillations. " Is not made with any justification, and seems to be an important concept for this paper. Would be good to unpack this for the reader and provide justification for this.

Thanks for pointing this out. We now added two references and expanded on the relationship between frequency, membrane potential and neural firing.

Manuscript lines 78-85: “One advantage of such a tighter temporal packing of spikes of an upstream neuron is that it is more likely to overcome the firing threshold of a downstream neuron because the individual spikes build on each other before the membrane potential fully drops back to baseline (20, 21). Therefore, a neural assembly which synchronizes firing at faster oscillations is more likely to drive a down-stream neural assembly compared to synchronization at slower oscillations. This may be the reason for why fast (~65 Hz), but not slow (~40 Hz), γ oscillations in rodents (22), and humans (23) have been demonstrated to reflect memory encoding processes.”

3. Line 425: This is a really nice interpretation of the findings of how excitability changes the frequency of the theta and γ oscillations and seems plausible. This reference (Lefebvre et al. 2015. "Stimulus Statistics Shape Oscillations in Nonlinear Recurrent Neural Networks." The Journal of Neuroscience: 35 (7): 2895-2903.) might also be a nice to that supports this.

Thanks for pointing out this very interesting modelling paper. We now included a citation to that paper.

Manuscript lines 411-412: “It is therefore conceivable, and consistent with computational modelling studies (42), that the frequency of theta is subject to modulation of the level of excitation or cognitive states which correlate with memory outcome.”

4. Line 695 & 725: Why were the frequency ranges of low (2-40), and high (40-80) defined as such. As well PPC spectrum is restricted to 3-13Hz, and 45-75Hz for γ. Although admittedly the precise ranges are always open to debate theta is typically considered 4-8Hz and γ 30-60, with high γ above 60. The choice of frequency range will likely change the statistics for example in Figure S4 then the stats are false discovery rate corrected. Can the authors justify the frequency ranges used for these analyses?

For analytical reasons, we split the frequency range into a higher and lower range because we used different parameter settings for wavelet analysis for the two frequency ranges in order to best capture their temporal dynamics and to deal with data volume (e.g. for the lower frequency range we used a frequency resolution of 1 Hz and a cycle length of 6, for the higher frequency range we used a resolution of 2 Hz and a cycle length of 12). The exact cut-off frequency of 40 Hz was chosen because of a previous paper from our group where we showed that the fast γ oscillation (between 50-80 Hz) in the hippocampus is specifically related to memory encoding. This is now explained in the methods section.

Manuscript lines 689-693: “We used these different parameter settings for the two frequency to best capture their temporal dynamics and to deal with data volume. The exact cut-off frequency of 40 Hz was chosen because of a previous paper from our group where we showed that the fast γ oscillation (between 50-80 Hz) in the hippocampus is specifically related to memory encoding (23).”

Concerning the PPC spectra in figures 2 and 3, we actually show the full range of frequencies from 40 – 80 Hz and 2-40 Hz. For the high frequency range the FDR correction was applied to the full frequency range (40-80 Hz). For the lower frequency range we indeed applied the FDR correction only for the 3-13 Hz frequency band, because we focused on theta oscillations. However, the difference in PPC in the high theta range between hits and misses also survives FDR correction when applied to the whole frequency range. The only statistical analysis where we restricted the frequency ranges was for the peak detection analysis, because we wanted to avoid peaks at the boarders of the filtering window.

For figure S4 (now Figure 1 —figure supplement 4) the whole frequency range was used, i.e. 40-80 Hz for the higher and 2-40 Hz for the lower frequency ranges.

5. Line 760: The text to describe PSI seems very similar to some of the text in Nolte, and would suggest revising a bit.

Thanks for point this out. We now changed to wording to avoid copying Nolte whilst still giving an accurate description of the method.

Manuscript lines 749-756: “To assess the direction of information between spikes and distally coupled LFPs the Phase Slope Index (PSI) was applied (56). The PSI is a frequency resolved measure to discern the direction of information flow between two neuronal regions or time series (A and B). Like many other directional coupling measures, the PSI assumes a time delay for a signal to travel from A to B. If the speed of travel is constant, then the phase difference between sender and recipient increases with frequency and a positive slope of the phase spectrum can be expected. Hence, a positive phase slope indicates that A is the sender and B is the receiver, and vice versa.”

6. 770: A minimum of 60 spikes is used here, and for SFC 30. Why the difference, and how are these justified.

We admit that this was a somewhat arbitrary choice that was taken to increase the signal-to-noise ratio of this particular analysis. Critically, the PSI relies on estimating the slope of the phase spectrum, therefore a higher number of observations would increase the sensitivity of this particular analysis (as opposed to SFC where the smoothness of the spectrum is of less importance). However, to be fully transparent we also repeated this analysis with a minimum number of 30 spikes and report these results in Figure 4 —figure supplement 2. The results show the same difference in PSI between hits and misses, however, the PSI for hits vs zero does not survive FDR correction anymore.

Manuscript lines 765-768: “Finally, a minimum of 60 spikes was imposed to allow for a meaningful analysis of directional information flow between the two signals (see Figure 4 —figure supplement 2 for a control analysis with a minimum of 30 spikes which obtained similar results to those shown in Figure 4).”

7. 798: CFC can also be artificially elevated as would be SFC by broad band power introduced by incomplete spike removal and/ or multiunit activity. Can the authors show that such an artifact does not underlie the CFC measures?

We do not think that such artifacts are the driving force behind our CFC results, first because our effects are band limited to the peak frequency of γ (+/- 5 Hz, see Figure 5D). If interpolation artifacts would be influencing the CFC results, then this would lead to a broadband coupling of high frequency power as the reviewer correctly notes. Secondly, to show that these artefacts are not present in our data we restricted the CFC analysis to channels that were ‘silent’ i.e. did not contain single or multi-unit activity. The results from this new analysis replicated our original findings and are reported in Figure 5 —figure supplement 3.

Manuscript lines 303-305: “Finally, to address concerns about possible broadband power artifacts introduced by spike interpolation we replicated the results by excluding high-frequency power providing channels with SUA/MUA activity (see Figure 5 —figure supplement 3).”

[Editors' note: further revisions were suggested prior to acceptance, as described below.]

The manuscript has been significantly improved and most of the queries posted by the reviewers have been addressed satisfactorily. There are some remaining issues that might still need to be addressed, as outlined below:– Both reviewers were concerned with the contaminations of spikes when analysing data from the same electrode. Those were dealt with very well with analysis considering 'silent' electrodes and also by removing those electrodes contributing spikes and LFPs. As such the issue has been dealt with, yet, it might be best to present the analysis in the full paper without those confounds and present in the supplement the full dataset.

We agree with the Editors that it would be good to show the spike-LFP coupling results for non-overlapping channels in the main manuscript. We have therefore included these results in the main figure and describe it in the main text. Changes appear in the new Figure 2 and in the Results section of the main manuscript. However, we decided to keep the results of the full dataset which benefits from the full statistical power (i.e. excluding non-overlapping channels removes a fifth of the data resulting in reduced statistical power), and because the results from the control analysis do show that spike interpolation had no bearing on the reported results.

Manuscript lines 173-181: “When calculating local spike-LFP coupling it is necessary to interpolate spikes to prevent high frequency artefacts (32), however, this interpolation can again potentially introduce artefacts and inflate spike-LFP coupling especially for channels where spikes are coupled to the LFP of that same channel (which were ~20% of the data). To address this issue we split the local spike-LFP coupling data into channel pairs where the spikes and LFPs were measured on the same microwire, and where spikes and LFPs were measured on different microwires (but on the same bundle of B-F electrodes; see Figure 2B, right). The PPC profile for both was highly similar, suggesting that spike interpolation did not artificially inflate the spike-LFP coupling.”

Manuscript lines 185-187: “A similar yet slightly weaker effect was also observed when excluding data where spikes and LFPs were measured on the same microwire (see Figure 2C, right; T-test; p_uncorr_<0.05).”

Manuscript lines 193-195: “Again, excluding data where spikes and LFPs were measured on the same microwire replicated this shift in peak frequency for hits compared to misses (t28 = 1.75; p=0.045; Cohen’s d = 0.32; Figure 2D – right).”

– Schneider et al. 2021. Neuron has provided an important alternative explanation to the findings of oscillations as a mechanism for inter areal communication. While the authors have toned down a causal interpretation of oscillations, it would be best to discuss in further length the paper by Schneider.

We agree and have included a full paragraph where we discuss the implications of the Schneider et al. paper for our findings in detail (changes appear in lines 456-468).

Manuscript lines 456-468: “The idea that synchronized inter-regional oscillations reflect effective communication has recently been questioned by a study showing that inter-regional phase synchronization can be a consequence rather than the cause of connectivity (47). Oscillatory activity in a local circuit will be reflected in postsynaptic activity (i.e. the LFP) of any area that it projects to. Consequently, giving rise to phase locking in the LFP between the two areas which highlights a weakness of LFP based connectivity measures and raises the need for additional methods to disambiguate between scenarios where oscillations establish communication between regions, and where they simply are a consequence thereof. To this end, we not only report a memory dependent shift from slower to faster frequencies in theta and γ spike-LFP coupling, but critically also report a memory dependent shift in spike-to-spike coupling, with hits showing earlier co-firing compared to misses. This finding is consistent with the idea that nested coupling of fast theta and γ oscillations enable efficient neural communication. However, whether this shift of co-firing lag is caused by a speed up of theta and γ oscillations remains an open question.”